# Qronos: Correcting the Past by Shaping the Future... in Post-Training Quantization

**Shihao Zhang**[*]
Department of Mathematics
University of California, San Diego
shz051@ucsd.edu

**Haoyu Zhang**[*]
Department of Mathematics
University of California, San Diego
haz053@ucsd.edu

**Ian Colbert**
Software Architecture
Advanced Micro Devices, Inc.
icolbert@amd.com

**Rayan Saab**
Department of Mathematics & HDSI
University of California, San Diego
rsaab@ucsd.edu

## Abstract

We introduce Qronos, a new post-training quantization algorithm that not only explicitly corrects errors due to both weight and activation quantization, but also corrects errors accumulated from previously quantized layers. Our iterative algorithm is based on an interpretable and disciplined optimization framework that surpasses existing data-driven approaches. At each step, Qronos alternates between error correction and diffusion via optimal update rules. Importantly, we prove that Qronos admits an equivalent formulation that significantly improves algorithmic efficiency; we use our discovery to reduce peak memory usage by $18\times$ on Llama3 8B, and our scaling analysis shows a speedup of up to $13.8\times$ for a single-layer microbenchmark. We demonstrate compatibility with existing transformation techniques such as Hadamard-based incoherence processing and weight-activation scaling equalization, among others. We evaluate Qronos using recent language models in the Llama3 and Qwen3 families; Qronos consistently outperforms previous state-of-the-art adaptive rounding methods when quantizing the weights, activations, and/or KV caches to 4 bits or fewer.

## 1 Introduction

Recent advances in post-training quantization (PTQ) have enabled the practical use of few-bit weights and activations for large language model (LLM) inference, typically by focusing on one or both aspects of the quantization pipeline, visualized in Figure 1. The first aspect involves modifying the weights and activations of a model to make them more amenable to quantization, often through transformations that exploit invariances within the compute graph. The second aspect more directly concerns the design of the quantization mapping itself. It involves using data to minimize quantization error by either calibrating the quantization grid, which is defined by a bit width, scaling factor, and zero point, or adaptively rounding the (potentially transformed) weights.

The latest innovations in PTQ, including Ashkboos et al. (2024); Liu et al. (2025), among many others, are skewed towards proposing and improving transformations that address the quantization challenges exacerbated in LLMs. These studies often only consider round-to-nearest (RTN) and OPTQ (Frantar et al., 2023), also known as GPTQ. Meanwhile, our work explicitly focuses on improving the rounding method while remaining compatible with these transformations.

**Contributions.** We introduce Qronos as a new scalable algorithm that not only explicitly corrects quantization error in both the weights and activations, but also residual quantization error coming from previously quantized layers. In contrast, OPTQ can only correct weight quantization error. We derive Qronos in a well-disciplined and mathematically interpretable form, then rigorously derive an equivalent efficient implementation (see Theorem 3.1) that significantly improves algorithm scaling

---

[*]Equal contribution.

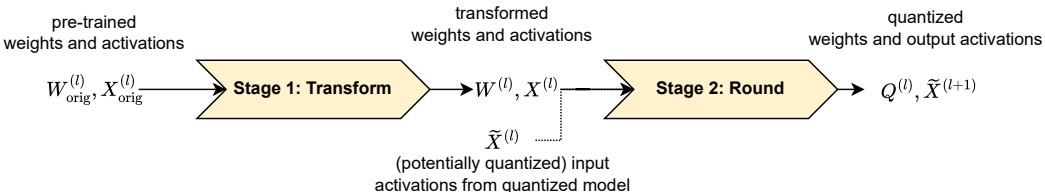

Figure 1: The modern quantization pipeline is typically a two-stage process consisting of (1) transformations that make weights and/or activations more amenable to quantization, followed by (2) rounding functions that map weights and/or activations onto a quantization grid.

(see Remark 3.3 and Section 4.3). As a non-trivial by-product, we address a theoretical blind spot of OPTQ by deriving a novel interpretation (Corollary 3.4), which shows that its seemingly local greedy update rules in fact correct the weight quantization error accumulated over all previous iterations. Our novel interpretation also offers clear geometric insights: at each step, OPTQ performs an optimal grid selection followed by an orthogonal projection onto a lower dimensional hyperplane spanned by future columns of the data matrix. This is one of the first results on the geometry of LLM quantization, among a few concurrent works (Birnick, 2025; Chen et al., 2025).

We evaluate Qronos on the Llama3 (Grattafiori et al., 2024) and Qwen3 (Yang et al., 2025) model families, and compare against RTN, OPTQ, GPFQ (Lybrand and Saab, 2021) and GPTAQ (Li et al., 2025) while demonstrating compatibility with notable transformations for both weight-only quantization and weight-activation quantization. To our knowledge, this is the first work to isolate the impact of the rounding algorithm through a carefully designed experimental setup that fixes the quantization grid for each transformation method (or lack thereof). Our experiments show that Qronos consistently yields marked improvement over existing methods, as highlighted in Table 1.

Table 1: **Weight-only quantization of Llama3 foundation models.** We jointly apply Hadamard-based incoherence processing (Ashkboos et al., 2024) and MagR (Zhang et al., 2024) as quantization transforms (stage 1 in Figure 1) and compare different rounding methods (stage 2).

|         |       | WikiText2 ($\downarrow$) | | | 0-shot ($\uparrow$) | | |
|---------|-------|------|------|------|------|------|------|
|         |       | 1B   | 3B   | 8B   | 1B   | 3B   | 8B   |
| BF16    | -     | 8.9  | 7.1  | 5.9  | 59.4 | 67.5 | 74.4 |
|         | RTN   | 3e3  | 5e3  | 3e3  | 32.4 | 32.2 | 33.0 |
|         | OPTQ  | 24.6 | 13.2 | 10.4 | 39.3 | 47.3 | 55.2 |
| 2-bit   | GPFQ  | 25.8 | 14.4 | 11.3 | 38.6 | 46.9 | 51.8 |
|         | GPTAQ | 22.0 | 12.2 | 9.6  | 39.8 | 49.2 | 54.8 |
|         | Qronos| **17.8** | **11.4** | **9.3** | **42.6** | **50.7** | **55.8** |
|         | RTN   | 5e5  | 4e4  | 9e4  | 32.3 | 32.9 | 32.2 |
|         | OPTQ  | 2e2  | 52.0 | 43.3 | 32.7 | 32.5 | 34.9 |
| 1.58-bit| GPFQ  | 1e2  | 51.3 | 35.8 | 32.4 | 32.6 | 33.4 |
|         | GPTAQ | 99.0 | 41.8 | 35.3 | 33.3 | 33.7 | 34.7 |
|         | Qronos| **39.3** | **22.8** | **18.0** | **34.8** | **36.5** | **37.8** |

## 2  BACKGROUND AND RELATED WORK

We first provide a short review of prior works that focus on the two key aspects of quantization we have mentioned: transformation techniques and rounding schemes. Figure 1 illustrates how these two aspects interact within the quantization pipeline.

**Methods based on transformations.** Many recent works propose transformations of weights and/or activations to facilitate quantization. One line of work, initially proposed for MobileNets (Nagel et al., 2019), exploits scaling invariance in neural network compute graphs to equalize the range or precision of weights and activations before quantization. Recent variants leverage scale invariance to redistribute quantization difficulty between weights and activations, with various proposals for learning scales or ranges based on custom objective functions (Xiao et al., 2023; Shao et al., 2024; Lin et al., 2024). Another line of work uses rotations within a compute graph to normalize weight

and activation distributions, initially leveraging random orthogonal rotations to promote weight incoherence (Chee et al., 2023). Recent variants employ efficient Hadamard rotations (Tseng et al., 2024; Ashkboos et al., 2024), Stiefel manifold optimizations (Liu et al., 2025; Hu et al., 2025), and rotation expansion techniques (Adepu et al., 2024; Franco et al., 2025a). Finally, distinct from these invariance-based approaches, MagR (Zhang et al., 2024) directly minimizes the $\ell_\infty$ norm of weights via proximal gradient descent to reduce dynamic range before quantization. While we do not introduce novel transformations of this type in this work, we demonstrate that existing transformations can be combined with our proposed method.

**Methods based on rounding.** The earliest line of work on rounding relies on continuous optimization strategies based on gradient descent (Nagel et al., 2020). Although more recent methods exist (Hubara et al., 2021; Li et al., 2021), they had not been commonly evaluated on LLMs due to their computational cost until Cheng et al. (2024). Thus, early work on LLMs focused on grid scaling or shifting to reduce weight quantization error; for example, LLM.int8() (Dettmers et al., 2022) and ZeroQuant (Yao et al., 2022) directly round to nearest after heuristically selecting the quantization grid (*i.e.*, bit width, scaling factors, and zero points). The most relevant line of work to ours adopts principled discrete optimization using greedy, gradient-free rounding strategies to select quantized weights to minimize the layer-wise reconstruction error, and includes OBQ (Frantar and Alistarh, 2022), OPTQ (Frantar et al., 2023), GPFQ (Lybrand and Saab, 2021; Zhang et al., 2023) and GPTAQ (Li et al., 2025). Qronos falls within this category.

**Notation.** Throughout the paper, the weight matrix of a layer is denoted by $W \in \mathbb{R}^{N \times N'}$, where each of the $N'$ columns represents a $N$-dimensional channel. $\mathcal{A}$ denotes the discrete quantization grid (or alphabet) used for weight quantization, and $\mathcal{Q}$ denotes the corresponding RTN operator associated with $\mathcal{A}$, given by $\mathcal{Q}(W) := s \cdot \left(\text{clip}\left(\left\lceil \frac{W}{s} \right\rceil + z; \min \mathcal{A}, \max \mathcal{A}\right) - z\right)$. Here, $\text{clip}(x; a_{\min}, a_{\max}) = \min\{\max\{x, a_{\min}\}, a_{\max}\}$, while the quantization step size (or scaling factor) is denoted by $s$ and the quantization grid is shifted by an offset denoted by $z$, often referred to as a zero point. We specify our selection of $s, z \in \mathbb{R}^{N'}$ for the various settings in Section 4. When quantizing $W$, we use $X \in \mathbb{R}^{m \times N}$ to denote the input calibration dataset of $m$ samples (*e.g.*, tokens) for the layer, resulting from the original pre-trained model, and $\widetilde{X} \in \mathbb{R}^{m \times N}$ to denote the input calibration dataset coming from the partially quantized model. Given a vector $v \in \mathbb{R}^n$, we use $v_i$ for its $i$-th entry, $v_{\geq j}$ for the subvector $(v_j, \ldots, v_n)^\top$, and we define $v_{\leq j}$ analogously. $\|v\|$ is the Euclidean norm of $v$. Given a matrix $A \in \mathbb{R}^{m \times n}$, we use $A_i$ to denote its $i$-th column. We use $A_{\geq j}$ to denote the submatrix $(A_j, \ldots, A_n)$. Similarly, $A_{\geq 2, \geq 2}$ denotes the submatrix of $A$ obtained by removing the first row and the first column. We use $\text{col}(A)$ to denote the column space of $A$. $P_A$ is the orthogonal projection onto $\text{col}(A)$, and $P_{A^\perp}$ the projection onto its orthogonal complement. Throughout this paper, all indices start from 1, following the standard mathematical convention.

**Layer-wise reconstruction and error correction.** Data-driven weight quantization methods typically aim to approximately minimize[1] the layer-wise reconstruction error given by

$$\min_{Q \in \mathcal{A}^{N \times N'}} \|XW - XQ\|_F^2. \tag{1}$$

At an arbitrary layer, the goal is to compute a quantized weight matrix $Q \in \mathcal{A}^{N \times N'}$ that preserves the output activations $XW$ under quantization. In practice, however, quantizing weights in earlier layers affects the input to subsequent layers. Let $\widetilde{X} \in \mathbb{R}^{m \times N}$ denote the activation matrix produced by a partially quantized model, where earlier layers have already been quantized. To account for the propagation of quantization error, we use a modified formulation, instead of Equation 1, that targets the mismatch between the original output $XW$ and $\widetilde{X}Q$ by approximately solving

$$\min_{Q \in \mathcal{A}^{N \times N'}} \|XW - \widetilde{X}Q\|_F^2. \tag{2}$$

The type of mismatch in this formulation is typically not addressed in the literature but arises naturally in both weight-only and weight-activation quantization settings. For instance, in weight-only quantization, $\widetilde{X}$ arises as the output of previously quantized layers, while in weight-activation quantization, one may encounter $\mathcal{Q}(\widetilde{X})$ rather than $\widetilde{X}$ if activations are quantized. Throughout this paper, we use the notation $(X, \widetilde{X})$ to refer generically to mismatched input pairs.

---

[1]Equation 1 is an instance of integer least-squares problems, which are known to be NP-hard (Hassibi and Vikalo, 2002). Thus, the best that one can hope for are approximate solutions.

## 3 QRONOS

We begin by describing the iterations associated with Qronos in Section 3.1. The iterations follow a disciplined and mathematically interpretable framework that alternates between error correction and diffusion using optimal update rules. We then prove that the explicit solutions to these minimization problems admit an efficient implementation. In Section 3.2, we provide deeper intuition behind Qronos in the context of previous state-of-the-art rounding algorithms, namely GPFQ and OPTQ. We also derive a novel interpretation of OPTQ (Corollary 3.4), which shows that it corrects the cumulative weight quantization error incurred over all the previous iterations. The proofs for all results in Section 3 are provided in the appendix.

### 3.1 ALGORITHM AND EFFICIENT IMPLEMENTATION

Let us first note that Qronos can process each column $w \in \mathbb{R}^N$ of $W \in \mathbb{R}^{N \times N'}$ independently and in parallel to produce each column $q \in \mathcal{A}^N$ of $Q \in \mathcal{A}^{N \times N'}$. Ideally, the goal is to find $q$ that minimizes $\frac{1}{2}\|Xw - \widetilde{X}q\|^2$. Since this problem is NP-hard, we propose an efficient sequential algorithm to approximate its solution. At each iteration, Qronos first selects the quantized weight that optimally corrects the current approximation error, holding the remaining weights fixed; see Equation 3 below. It then updates the unquantized weights to optimally compensate for the rounding error, a process we refer to as error diffusion; see Equation 4.

Let $w$, without superscripts or subscripts, denote the original unquantized weights. After determining $q_{t-1}$, let $w_{\geq t}^{(t-1)}$ represent the updated unquantized weights corresponding to indices $t$ through $N$. The full state of the algorithm after step $t-1$ is thus given by the vector $w^{(t-1)} = (q_{\leq t-1}, w_{\geq t}^{(t-1)})$, with the initialization $w^{(0)} = w$. At step $t$, the algorithm alternates between selecting $q_t$ through error correction and updating the remaining weights through error diffusion. The update rules are given by

$$q_t = \underset{p \in \mathcal{A}}{\operatorname{argmin}} \frac{1}{2}\|Xw - \sum_{j=1}^{t-1} q_j \widetilde{X}_j - p\widetilde{X}_t - \sum_{j=t+1}^{N} w_j^{(t-1)} \widetilde{X}_j\|^2, \tag{3}$$

$$w_{\geq t+1}^{(t)} = \underset{(v_{t+1},\dots,v_N) \in \mathbb{R}^{N-t}}{\operatorname{argmin}} \frac{1}{2}\|Xw - \sum_{j=1}^{t} q_j \widetilde{X}_j - \sum_{j=t+1}^{N} v_j \widetilde{X}_j\|^2. \tag{4}$$

These optimization problems admit the following closed-form solutions (see Proposition E.1):

$$q_t = \mathcal{Q}\left(\frac{\left\langle Xw - \sum_{j=1}^{t-1} q_j \widetilde{X}_j - \sum_{j=t+1}^{N} w_j^{(t-1)} \widetilde{X}_j, \widetilde{X}_t \right\rangle}{\|\widetilde{X}_t\|^2}\right), \tag{5}$$

$$w_{\geq t+1}^{(t)} = \widetilde{X}_{\geq t+1}^{\dagger}\left(Xw - \widetilde{X}_{\leq t} q_{\leq t}\right). \tag{6}$$

While these expressions follow directly from the optimization problems, computing $q_t$ and $w_{\geq t+1}^{(t)}$ in this form is not computationally efficient and scales poorly, as we will show in Section 4.3. To address this, we present Theorem 3.1, which shows that for all $t \geq 2$, $q_t$ can be computed via RTN, enabling a simpler implementation. In Lemma 3.2, we further show that the update for $w_{\geq t+1}^{(t)}$ also admits an efficient implementation using Cholesky decomposition to solve the associated least-squares problem. Together, these results yield a practical and scalable implementation of Qronos.

**Theorem 3.1.** *Let $(q_t, w_{\geq t}^{(t-1)})$ be the iterates generated by Equation 3 and Equation 4, with initialization $w_{\geq 1}^{(0)} = w$. Define an alternative sequence $(\hat{q}_t, \hat{w}_{\geq t}^{(t-1)})$ using the same initialization*

$\hat{w}_{\geq 1}^{(0)} = w$, *by setting*

$$\hat{q}_1 = \arg\min_{p \in \mathcal{A}} \frac{1}{2}\|Xw - p\widetilde{X}_1 - \sum_{j=2}^{N} w_j \widetilde{X}_j\|^2, \tag{7}$$

$$\hat{w}_{\geq 2}^{(1)} = \arg\min_{(v_2,\dots,v_N) \in \mathbb{R}^{N-1}} \frac{1}{2}\|Xw - \hat{q}_1\widetilde{X}_1 - \sum_{j=2}^{N} v_j \widetilde{X}_j\|^2, \tag{8}$$

*and, for $t = 2, \dots, N$, define*

$$\hat{q}_t = \mathcal{Q}(\hat{w}_t^{(t-1)}), \tag{9}$$

$$\hat{w}_{\geq t+1}^{(t)} = \arg\min_{(v_{t+1},\dots,v_N) \in \mathbb{R}^{N-t}} \frac{1}{2}\|(\hat{q}_t - \hat{w}_t^{(t-1)})\widetilde{X}_t + \sum_{j=t+1}^{N} (v_j - \hat{w}_j^{(t-1)})\widetilde{X}_j\|^2. \tag{10}$$

*Then for $t = 1, \dots, N$, the two procedures yield identical iterates: $(q_t, w_{\geq t}^{(t-1)}) = (\hat{q}_t, \hat{w}_{\geq t}^{(t-1)})$.*

Starting from the second iteration, Theorem 3.1 shows that the updates in Equation 3 and Equation 4 can be equivalently reformulated as Equation 9 and Equation 10, respectively. This reformulation allows $q_t$ to be obtained via RTN for $t \geq 2$, followed by an adjustment of the remaining weights using only the (potentially quantized) activation matrix $\widetilde{X}$ to compensate for the one-step quantization error $(q_t - w_t^{(t-1)})\widetilde{X}_t$.

To further accelerate this adjustment step, we now present Lemma 3.2, which establishes the equivalence of the update in Equation 10 (for $t \geq 2$) with a Cholesky-based least-squares solution[2]. For notational simplicity, we slightly abuse the indexing by treating $t = 2$ as a 'restart.'

**Lemma 3.2** (Equivalence of Least-Squares Formulation and Cholesky Formulation). *Assume that $H = X^\top X$ is invertible, and let $H^{-1} = LL^\top$ denote its Cholesky decomposition, with $L$ lower triangular. Then, starting from $w^{(0)} = w$, the update rules*

$$q_t = \mathcal{Q}(w_t^{(t-1)}), \tag{11}$$

$$w_{\geq t+1}^{(t)} = \arg\min_{(v_{t+1},\dots,v_N) \in \mathbb{R}^{N-t}} \frac{1}{2}\|(q_t - w_t^{(t-1)})X_t + \sum_{j=t+1}^{N} (v_j - w_j^{(t-1)})X_j\|^2 \tag{12}$$

*are equivalent to the Cholesky-based iterations*

$$q_t = \mathcal{Q}(w_t^{(t-1)}), \tag{13}$$

$$w_{\geq t+1}^{(t)} = w_{\geq t+1}^{(t-1)} + \Delta^{(t)}, \tag{14}$$

*where $\Delta^{(t)} = -(w_t^{(t-1)} - q_t)\frac{L_{\geq t+1,\,t}}{L_{tt}} \in \mathbb{R}^{N-t}$.*

**Remark 3.3** (**Memory Efficiency**). *At the first iteration, both $q_1$ and $w_{\geq 2}^{(1)}$ depend on $\widetilde{X}, X \in \mathbb{R}^{m \times N}$, requiring $\mathcal{O}(mN)$ peak memory, often where $m \gg N$. For example, Llama3.1-8B requires over 30 GB just to store 128 samples of 2048-token sequences at* `float32`*. We optimize this first iteration to use only square matrices such that*

$$q_1 = \mathcal{Q}\left(\frac{G_{1,\geq 1}w - H_{1,\geq 2}w_{\geq 2}^{(0)}}{H_{11}}\right), \tag{15}$$

$$w_{\geq 2}^{(1)} = (H_{\geq 2,\geq 2})^{-1}\left(G_{\geq 2,\geq 1}w - H_{\geq 2,1}q_1\right), \tag{16}$$

*where $G = \widetilde{X}^T X \in \mathbb{R}^{N \times N}$ and $H = \widetilde{X}^T \widetilde{X} \in \mathbb{R}^{N \times N}$; see Proposition E.2 for a justification. Note that calculating $G$ and $H$ does not require storing $\widetilde{X}, X$, as one can sequentially accumulate the outer products of each of the $m$ samples. Thus, this square matrix formulation reduces peak memory requirements of Qronos from $\mathcal{O}(mN)$ to $\mathcal{O}(N^2)$, yielding an $\mathbf{18\times}$ reduction in the case of Llama3.1-8B. We note that Colbert et al. (2024) similarly identify a memory optimization for GPFQ, but use singular value decompositions that may not scale well with $N$.*

---

[2]We do not claim that Lemma 3.2 is novel, though we were unable to find it stated explicitly in the literature.

This completes our reduction of the original updates (Equations 3 and 4) to the equivalent implementation given by Equations 13, 14, 15, and 16. The pseudocode for this efficient version is provided in Appendix A. We further present a runtime analysis comparing this efficient version with the base version (*i.e.*, a direct evaluation of the closed-form solution) in Section 4.3.

## 3.2 THEORETICAL INTERPRETATION AND INTUITION

Theorem 3.1 and Lemma 3.2 connect the initial disciplined optimization formulation of Qronos to our efficient implementation. These results guarantee that Qronos is both interpretable and scalable, explicitly correcting error from the mismatched input pairs $X$ and $\widetilde{X}$. Here, we provide deeper intuition in the context of previous state-of-the-art rounding algorithms, namely GPFQ and OPTQ.

When quantizing $w$, GPFQ (Lybrand and Saab, 2021; Zhang et al., 2023; Zhang and Saab, 2023) interprets $Xw$ as the endpoint of the path $\sum_{j=1}^{t} w_j X_j$ for $t = 1, ..., N$, and handles mismatched inputs by aiming to match $\sum_{j=1}^{t} w_j X_j$ and $\sum_{j=1}^{t} q_j \widetilde{X}_j$ for all $t$. More precisely, $q_t$ is selected as $\arg\min_{p \in \mathcal{A}} \| \sum_{j=1}^{t} w_j X_j - \sum_{j=1}^{t-1} q_j \widetilde{X}_j - p\widetilde{X}_t \|^2$.

Although path following handles the case when $X = \widetilde{X}$ well, additional considerations are required when $X \neq \widetilde{X}$ since, in such a case, the tails of the two paths generally do not align when $\sum_{i=t+1}^{N} w_i(X_i - \widetilde{X}_i) \neq 0$. Qronos handles this drawback by adopting a natural remedy to replace the unquantized weights $w_i$ by auxiliary weights $w_i^{(t)}$, for $i \geq t + 1$, so that

$$\sum_{i=1}^{t} q_i \widetilde{X}_i + \sum_{i=t+1}^{N} w_i^{(t)} \widetilde{X}_i \approx Xw = \sum_{i=1}^{N} w_i X_i.$$

OPTQ (Frantar et al., 2023) explores a similar weight update idea, but only in the case where $X = \widetilde{X}$, by modifying the remaining unquantized weights after $q_t$ is selected. The Cholesky reformulation used in Lemma 3.2 also resembles the key mechanism in OPTQ. In this way, the runtime of Qronos scales similarly to OPTQ while also explicitly addressing the mismatch between $X$ and $\widetilde{X}$; see Section 4.3 for details. This unexpected connection of Qronos to OPTQ also allows us to derive a novel interpretation of OPTQ, which we now present.

**Corollary 3.4.** *The OPTQ iterations, when applied to a single layer input $X$, are equivalent to*

$$q_t = \arg\min_{p \in \mathcal{A}} \frac{1}{2} \| Xw - \sum_{j=1}^{t-1} q_j X_j - pX_t - \sum_{j=t+1}^{N} w_j^{(t-1)} X_j \|^2, \tag{17}$$

$$w_{\geq t+1}^{(t)} = \arg\min_{(v_{t+1}, ..., v_N) \in \mathbb{R}^{N-t}} \frac{1}{2} \| Xw - \sum_{j=1}^{t} q_j X_j - \sum_{j=t+1}^{N} v_j X_j \|^2, \tag{18}$$

*with $w_{\geq 1}^{(0)} = w$.*

In other words, the updated weights and quantized weights at every iteration $t$ that are produced by OPTQ are identical to those produced by Equations 17 and 18. In particular, Equation 18 shows that, at each step the updated weights $w_{\geq t+1}^{(t)}$ indeed optimally correct for the errors produced by the hitherto quantized sequence $q_1, ..., q_t$ via orthogonal projection onto $\text{col}(X_{\geq t+1})$, as further discussed in Appendix H.

Noticeably, OPTQ suffers from a systematic bias when the activation mismatch is non-negligible as, unlike Qronos, it does not explicitly minimize the true discrepancy $\min_{q \in \mathcal{A}^N} \| Xw - \widetilde{X}q \|_2$. Consequently, as discussed in Appendix D, Qronos consistently reduces the relative error (measured in $\ell_2$ norm) of block outputs compared to OPTQ, as illustrated in Figure 3.

## 4 EXPERIMENTS

The core contribution of this work is Qronos—our principled data-driven rounding algorithm that alternates between (1) explicitly correcting quantization error due to both the weights and activations,

Table 2: **2-bit weight-only quantization of Qwen3 instruction fine-tuned models.** We apply HIP (stage 1 in Figure 1) and compare different rounding methods (stage 2).

| | WikiText2 ($\downarrow$) | | | | | | 0-shot ($\uparrow$) | | | | | |
|---|---|---|---|---|---|---|---|---|---|---|---|---|
| | 0.6B | 1.7B | 4B | 8B | 14B | 32B | 0.6B | 1.7B | 4B | 8B | 14B | 32B |
| BF16 | 18.6 | 15.2 | 12.2 | 8.6 | 7.6 | 6.8 | 51.1 | 61.4 | 68.9 | 72.4 | 75.4 | 77.2 |
| RTN | 7e5 | 8e6 | 4e5 | 4e4 | 3e5 | 1e5 | 32.1 | 31.9 | 32.4 | 31.8 | 32.9 | 32.8 |
| OPTQ | 1e2 | 60.0 | 22.8 | 14.7 | 14.9 | 12.8 | 32.0 | 32.8 | 37.4 | 41.4 | 42.5 | 47.0 |
| GPFQ | 1e2 | 45.3 | 25.4 | 17.1 | 15.6 | 13.4 | 33.0 | 32.4 | 35.9 | 39.4 | 40.4 | 46.0 |
| GPTAQ | 74.5 | 37.0 | 21.0 | 13.6 | 14.4 | 12.9 | 32.3 | 34.0 | 38.7 | 42.5 | 43.3 | 47.3 |
| Qronos | **46.0** | **23.5** | **17.8** | **12.9** | **13.4** | **12.0** | **35.0** | **36.7** | **41.5** | **44.7** | **45.2** | **48.0** |

Table 3: **Weight-only quantization of Llama3 foundation models.** We individually apply various quantization transforms (stage 1 in Figure 1) to isolate the impact of different rounding functions (stage 2) when quantizing to 3 and 4 bits, respectively denoted W3 and W4.

| | | W3 | | | | | | W4 | | | | | |
|---|---|---|---|---|---|---|---|---|---|---|---|---|---|
| | | WikiText2 ($\downarrow$) | | | 0-shot ($\uparrow$) | | | WikiText2 ($\downarrow$) | | | 0-shot ($\uparrow$) | | |
| Stage 1 | Stage 2 | 1B | 3B | 8B | 1B | 3B | 8B | 1B | 3B | 8B | 1B | 3B | 8B |
| BF16 | - | 8.9 | 7.1 | 5.9 | 59.4 | 67.5 | 74.4 | 8.9 | 7.1 | 5.9 | 59.4 | 67.5 | 74.4 |
| **None** | RTN | 2e4 | 1e4 | 3e4 | 32.3 | 32.4 | 32.6 | 18.0 | 10.1 | 8.4 | 49.1 | 60.8 | 67.4 |
| | OPTQ | 42.5 | 13.8 | 11.4 | 37.5 | 48.1 | 53.8 | 10.4 | 7.8 | 6.5 | 54.3 | 63.4 | 71.0 |
| | GPFQ | 35.3 | 13.4 | 11.1 | 35.7 | 49.9 | 53.5 | 10.4 | 7.8 | 6.5 | 56.0 | **65.2** | 71.2 |
| | GPTAQ | 28.4 | 12.6 | 10.3 | 39.3 | 49.6 | **57.1** | 10.3 | 7.8 | 6.5 | **56.3** | 63.3 | 71.0 |
| | Qronos | **22.8** | **11.3** | **9.3** | **39.5** | **53.1** | 56.7 | **10.1** | **7.6** | **6.4** | 56.2 | 64.5 | **72.0** |
| **Smooth Quant** | RTN | 6e3 | 9e3 | 5e4 | 32.7 | 32.9 | 31.4 | 15.2 | 9.6 | 8.1 | 51.4 | 61.5 | 67.5 |
| | OPTQ | 29.6 | 13.6 | 12.6 | 37.0 | 46.9 | 47.3 | 10.4 | 7.9 | 6.6 | 56.2 | **65.3** | 70.2 |
| | GPFQ | 30.1 | 14.7 | 12.9 | 36.5 | 44.8 | 45.4 | 10.8 | 7.9 | 6.7 | 53.9 | 64.4 | 69.9 |
| | GPTAQ | 25.0 | 12.9 | 11.4 | 37.9 | 46.8 | 49.1 | 10.4 | 7.9 | 6.6 | 55.2 | 63.1 | **71.2** |
| | Qronos | **19.1** | **11.6** | **10.3** | **40.7** | **50.6** | 50.5 | **10.3** | **7.8** | **6.5** | **56.7** | 64.8 | 70.2 |
| **MagR** | RTN | 2e3 | 2e3 | 5e4 | 33.8 | 33.5 | 35.1 | 13.8 | 10.3 | 7.2 | 53.1 | 58.1 | 69.7 |
| | OPTQ | 20.1 | 12.9 | 8.1 | 44.2 | 45.6 | 59.7 | 10.3 | **8.0** | 6.5 | **56.4** | 60.0 | 69.0 |
| | GPFQ | 21.0 | 14.0 | 8.3 | 43.9 | 48.4 | **61.7** | 10.4 | **8.0** | 6.5 | 55.4 | **61.1** | 70.3 |
| | GPTAQ | 18.0 | 12.4 | 8.0 | **46.8** | **51.2** | 60.7 | 10.3 | **8.0** | **6.4** | 56.2 | 60.0 | 70.3 |
| | Qronos | **16.9** | **11.8** | **7.8** | 46.6 | **51.2** | 60.0 | **10.1** | **8.0** | **6.4** | 56.2 | **61.1** | **70.4** |
| **HIP** | RTN | 7e2 | 3e2 | 1e2 | 34.2 | 33.3 | 36.3 | 13.8 | 8.8 | 7.2 | 52.0 | 62.8 | 70.0 |
| | OPTQ | 16.1 | 10.3 | 8.6 | 44.1 | 56.6 | 58.8 | 9.9 | 7.6 | 6.3 | 56.8 | **66.1** | 72.1 |
| | GPFQ | 16.6 | 10.4 | 8.6 | 44.9 | 54.8 | 58.9 | 9.9 | 7.6 | 6.3 | 56.5 | 65.7 | 72.0 |
| | GPTAQ | 14.7 | 9.9 | 8.3 | 46.5 | 56.9 | 59.3 | 9.8 | **7.5** | 6.3 | **57.8** | 66.0 | **72.4** |
| | Qronos | **12.9** | **9.3** | **7.8** | **48.1** | **59.6** | **62.2** | **9.6** | **7.5** | **6.2** | 57.1 | 65.9 | 71.0 |

and (2) diffusing excess error into future weights yet to be quantized. Thus, our primary comparison metric is preserving model quality in challenging quantization scenarios. We design our experiments to isolate the impact of the rounding function (stage 2 in Figure 1), while varying the quantization transforms (stage 1 in Figure 1), as further discussed in Sections 4.1 and 4.2.

**Models & Datasets.** We conduct experiments on Llama3 (Grattafiori et al., 2024) and Qwen3 (Yang et al., 2025) models using WikiText2 (Merity et al., 2016) for evaluation. We use, without modification, the implementations made publicly available via Huggingface (Wolf et al., 2020). We provide additional results in Appendix B. We use LightEval (Fourrier et al., 2023) to evaluate generalization via 5 zero-shot reasoning tasks: ARC (challenge and easy) (Clark et al., 2018), HellaSwag (Zellers et al., 2019), PIQA (Bisk et al., 2020), and Winogrande (Sakaguchi et al., 2021), and report the normalized average accuracy.

**Setup.** We implement Qronos in PyTorch (Paszke et al., 2019) using the Brevitas quantization library (Franco et al., 2025b), and quantize all models using a single AMD MI210 GPU with 64 GB of memory. Unless otherwise specified, we construct our calibration dataset using 128 random sequences of 2048 tokens sampled from the WikiText2 dataset for all data-driven PTQ algorithms. We

Table 4: **Weight-activation quantization of Llama3 foundation models.** We individually apply various transformations (stage 1) to isolate the impact of different rounding functions (stage 2).

| | | W4A4KV16 | | | | | | W4A4KV4 | | | | | |
| | | WikiText2 ($\downarrow$) | | | 0-shot ($\uparrow$) | | | WikiText2 ($\downarrow$) | | | 0-shot ($\uparrow$) | | |
| Stage 1 | Stage 2 | 1B | 3B | 8B | 1B | 3B | 8B | 1B | 3B | 8B | 1B | 3B | 8B |
|---|---|---|---|---|---|---|---|---|---|---|---|---|---|
| BF16 | - | 8.9 | 7.1 | 5.9 | 59.4 | 67.5 | 74.4 | 8.9 | 7.1 | 5.9 | 59.4 | 67.5 | 74.4 |
| **QuaRot** | RTN | 22.0 | 12.6 | 9.6 | 45.4 | 55.0 | 62.6 | 41.8 | 22.0 | 15.9 | 41.5 | 49.8 | 57.4 |
| | OPTQ | 14.3 | 9.8 | 8.0 | 50.4 | 59.9 | 66.7 | 19.8 | 14.3 | 10.3 | 45.8 | 56.2 | 64.1 |
| | GPFQ | 13.6 | 9.3 | 7.6 | 50.9 | 60.9 | 67.6 | 22.0 | 14.7 | 11.4 | 43.3 | 53.9 | 59.8 |
| | GPTAQ | 13.4 | 9.2 | **7.4** | **51.2** | 61.4 | 68.1 | 18.0 | 12.2 | **9.3** | 46.6 | **57.3** | **64.8** |
| | Qronos | **13.2** | **9.1** | **7.4** | 50.9 | **61.5** | **68.9** | **17.8** | **11.6** | **9.3** | **47.8** | 57.3 | **64.8** |
| **SmoothRot** | RTN | 22.4 | 12.2 | 11.1 | 42.9 | 54.8 | 62.6 | 39.3 | 19.5 | 34.3 | 40.4 | 49.4 | 50.6 |
| | OPTQ | 13.6 | 9.5 | 7.9 | 51.0 | 60.3 | 68.5 | 18.6 | 12.9 | 16.1 | 45.9 | 55.9 | 59.1 |
| | GPFQ | 12.9 | **8.8** | 7.4 | 50.4 | **62.0** | 67.7 | 20.8 | 14.3 | 12.2 | 44.4 | 54.9 | 59.0 |
| | GPTAQ | **12.6** | 8.9 | 7.3 | **51.1** | 61.4 | 68.8 | **16.6** | **11.6** | 10.8 | **48.6** | **57.8** | 63.7 |
| | Qronos | **12.6** | **8.8** | **7.2** | 50.8 | 60.9 | **69.4** | 16.9 | **11.6** | **9.5** | 47.1 | **57.8** | **65.2** |
| **SpinQuant** | RTN | 20.5 | 12.6 | 9.3 | 47.7 | 57.5 | 64.2 | 33.5 | 20.2 | 13.4 | 43.1 | 52.2 | 60.8 |
| | OPTQ | 13.4 | 9.2 | 7.7 | 52.0 | 61.1 | 67.0 | 17.9 | 15.0 | 8.9 | 47.9 | **58.5** | 65.5 |
| | GPFQ | 13.5 | 9.2 | 7.5 | 51.2 | 61.2 | 67.0 | 21.1 | 14.3 | 10.9 | 45.3 | 53.6 | 60.9 |
| | GPTAQ | 12.9 | 9.0 | 7.4 | 51.8 | 61.1 | 68.3 | 17.1 | NaN | **8.7** | **49.4** | NaN | 65.3 |
| | Qronos | **12.3** | **8.7** | **7.2** | **52.8** | **62.1** | **68.4** | **16.4** | **11.1** | **8.7** | 48.2 | 58.2 | **65.8** |

compare Qronos against RTN and the unmodified Brevitas implementations of OPTQ and GPFQ, also leveraging the unmodified Brevitas implementations of the various quantization transforms. We provide quantization transform hyperparameter details in Appendix C, as well as ablation studies.

**Baselines.** Our baselines are RTN, OPTQ, GPFQ and GPTAQ. For OPTQ, we use the standard dampened covariance matrix $\widetilde{H} = H + \lambda I$, where $\lambda$ is 1% of the average diagonal of $H$. We similarly use a dampened covariance matrix for Qronos, but choose $\lambda$ to be based on the maximum singular value of $H$ such that $\lambda = \alpha \cdot \sigma_1$, which limits the condition number of $\widetilde{H}$ to be less than $\alpha^{-1}$. We select $\alpha = 1e^{-6}$ for weight-only quantization and $\alpha = 1e^{-3}$ for weight-activation quantization. Additionally, we apply GPFQ, GPTAQ, and Qronos block-by-block; this corresponds to resetting $\widetilde{X} = X$ at the beginning of each block. Finally, we quantize weights in descending order of the diagonals of $H$, as is now common practice (IST-DASLab, 2022; Franco et al., 2025b).

## 4.1 WEIGHT-ONLY QUANTIZATION

We first present state-of-the-art 2-bit and 1.58-bit results for weight-only PTQ on Llama3 foundation models, controlling for the quantization transform and grid selection while varying the rounding function. We quantize weights using the standard asymmetric weight quantizer (Frantar et al., 2023), where scaling factor $s$ and zero point $z$ are defined per-channel on a scaled min-max grid such that $s = \beta \cdot (\max(w) - \min(w))/(2^b - 1)$ and $z = \beta \cdot \min(w)/s$. Following the analysis of Zhang et al. (2024), we choose $\beta = 0.8$ when quantizing to 2 bits or fewer. We combine Hadamard-based incoherence processing (HIP) (Tseng et al., 2024; Ashkboos et al., 2024) with weight magnitude reduction (MagR) (Zhang et al., 2024) to jointly act as our quantization transform, as they are both known to be effective at few-bit weight quantization (Chee et al., 2023; Adepu et al., 2024). We present our results in Table 1, as well as the BF16 baselines, and highlight that Qronos consistently outperforms existing rounding methods. For example, when compared to OPTQ, Qronos provides a $1.4\times$ reduction in WikiText2 perplexity and $+3.3\%$ increase in average zero-shot accuracy for Llama3.2-1B at 2 bits, and a massive improvement in perplexity ($4.9\times$) at 1.58 bits. We provide additional 2-bit and 1.58-bit results with $\beta = 1$ in Appendix C.1.

Next, we present state-of-the-art 2-bit weight-only PTQ results on Qwen3 instruction fine-tuned models. Here, we use HIP as our quantization transform then tune the grid to minimize the mean squared error loss between the transformed weights and their RTN-quantized counterparts via a linear search over $s$ and $z$. Table 2 provides the results from Qwen3 0.6B to 32B. Qronos again yields clear and consistent improvements for all models in this family.

Table 5: **Calibration Runtime Analysis.** We report the end-to-end calibration time of OPTQ and Qronos for the Qwen3 model family, normalized to Qwen3-0.6B, as measured on an AMD MI325X.

|  | 0.6B | 1.7B | 4B | 8B | 14B | 32B |
|---|---|---|---|---|---|---|
| OPTQ | 1.0 | 1.4 | 2.8 | 3.7 | 5.6 | 10.6 |
| Qronos | 1.2 | 1.6 | 3.1 | 4.0 | 6.1 | 11.5 |
| **Overhead** | 19.7% | 16.2% | 11.1% | 9.0% | 8.7% | 8.7% |

Finally, we present 3-bit and 4-bit weight-only PTQ results (denoted W3 and W4, respectively) on Llama3 foundation models while independently demonstrating compatibility with 3 notable quantization transforms: SmoothQuant (Xiao et al., 2023), MagR, and HIP. Table 3 shows the results across three models in the Llama3 family. For both W3 and W4, we use $\beta = 1$. Qronos consistently provides higher quality quantized models than RTN, OPTQ, GPFQ and GPTAQ, as measured in both WikiText2 perplexity and average zero-shot accuracy. Consistent with emerging work on rotation-based quantization transforms (Chee et al., 2023; Tseng et al., 2024), incoherence processing outperforms other transforms, with HIP + Qronos providing the best overall results. Note that HIP + OPTQ is similar in spirit to QuIP by Theorem 6 in (Chee et al., 2023), which equates LDLQ to OPTQ, with a notable difference that QuIP proposed random orthogonal matrices instead of Hadamard matrices.

## 4.2 Weight-Activation Quantization

We present 4-bit weight-activation quantization results with and without 4-bit KV cache quantization (denoted W4A4KV16 and W4A4KV4, respectively) while demonstrating compatibility with QuaRot (Ashkboos et al., 2024), SmoothRot (Czakó et al., 2025), and SpinQuant (Liu et al., 2025). We quantize weights using the standard symmetric weight quantizer with per-channel scaling factors optimized via linear search over the mean square error loss between the full-precision and quantized weights. We quantize activations using the standard asymmetric activation quantizer with dynamic per-token scaling factors and zero points defined on the min-max grid, as is common practice (Liu et al., 2025). When quantizing KV caches, we similarly use per-token scaling and zero points.

Table 4 shows the results across three foundation models in the Llama3 family. Qronos again consistently outperforms RTN, OPTQ, GPFQ and GPTAQ[3] as measured in both WikiText2 perplexity and average zero-shot accuracy. Consistent with emerging work on learned rotations (Liu et al., 2025; Hu et al., 2025; Franco et al., 2025a), SpinQuant outperforms QuaRot and SmoothRot, with SpinQuant + Qronos providing the best overall results with and without KV cache quantization. We remark that our experiments use per-token quantization for both the activations and KV caches, while Ashkboos et al. (2024) and Liu et al. (2025) both use per-group scaling for KV cache quantization.

Our experimental analysis reveals an important pattern: Qronos provides larger improvements as quantization tasks become more challenging. Specifically, Qronos demonstrates larger relative improvements over existing methods when transitioning from weight-only to weight-activation quantization (*i.e.,* W4 versus W4A4), and even more substantial gains when incorporating KV cache quantization (*i.e.,* W4A4 versus W4A4KV4). We further validate this pattern with additional W3A3 results in Appendix B (Table 8), which show larger improvements than both W4A4 and W3 weight-only quantization. These findings suggest that Qronos is particularly effective in scenarios where multiple sources of quantization error interact, making it especially valuable for aggressive quantization settings where traditional methods struggle to maintain model quality.

## 4.3 Hardware Efficiency and Runtime Analysis

The hardware efficiency benefits of quantization (*i.e.*, improved throughput, memory, power, and area) are well-established (Jacob et al., 2018; Colbert et al., 2024). Since Qronos and other rounding algorithms leave the compute graph unaltered, they capture these benefits without introducing inference overhead beyond the quantization transform. Prior works have already profiled inference

---

[3]We observed instability with GPTAQ, as reflected by the NaN entries in Table 4, and similar issues have been reported by others attempting to reproduce results from Li et al. (2025) with their official repository.

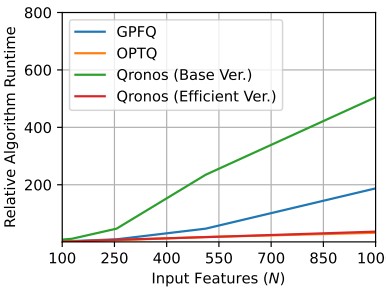 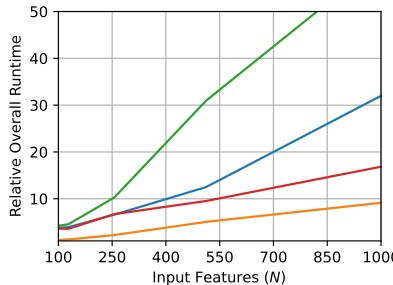

(a) Runtime of Rounding Algorithm     (b) Runtime of Quantization Pipeline

Figure 2: We compare the runtime of (a) the rounding algorithm and (b) the overall quantization pipeline as we scale the input features $N$, as measured on an AMD MI210. We average all measurements over 3 seeds and normalize to the runtime of OPTQ where $N = 32$.

speedups and overheads; for example, Ashkboos et al. (2024) report up to 2.16× speedup for W4A4 Llama2 7B over FP16, with Hadamard transforms adding at most 7% overhead. Therefore, we focus our runtime analysis on the quantization pipeline itself.

**Microbenchmark.** We perform our initial runtime analysis using a single linear layer. We use a calibration set of $m =$10,000 random data sampled from a $K$-dimensional Gaussian distribution. The linear layer has $K \in [32, 1024]$ inputs with $K/4$ outputs. Figure 2 shows how the runtime of OPTQ, GPFQ, and Qronos scale with $K$, where (a) isolates the algorithm runtime (i.e., without the added inference cost of calculating $H$ and $G$) and (b) aggregates the end-to-end runtime of calibration. To highlight the benefits of our equivalent formulation, we implement a base version of Qronos that uses the iterates for $q_t$ and $w_{\geq t+1}^{(t)}$ from Equations 5 and 6. Note that via Theorem 3.1 and Lemma 3.2, we significantly improve the runtime scaling of Qronos over the base version, with a **13.8×** reduction in algorithm runtime and a **3.6×** reduction in overall runtime when $K = 1024$.

**Calibration Runtime.** Compared with OPTQ, which only needs to collect $X$, GPFQ and Qronos require collecting both $\widetilde{X}$ and $X$ at each layer, which requires two forward passes (with and without quantization) and increases the overall quantization pipeline runtime. To evaluate the overhead of two forward passes in practice, we compare the calibration runtime of OPTQ and Qronos when quantizing the Qwen3 model family. Table 5 provides the runtimes for each model from 0.6B to 32B, normalized to the calibration runtime when using OPTQ to quantize Qwen3-0.6B on an AMD MI325X. We observe the overhead of Qronos decreases from 19.7% to 8.7% as model size increases from 0.6B to 32B, indicating that algorithmic cost dominates the cost of executing inference twice and underscoring the importance of Theorem 3.1.

## 5 CONCLUSIONS

We introduce Qronos—a new backpropagation-free rounding algorithm that alternates between correcting quantization error in both the weights and activations of previous layers and diffusing error into future weights within the current layer. Qronos is based on an interpretable and disciplined optimization framework, and it demonstrably surpasses existing data-driven approaches. Our implementation exploits several optimizations that together yield orders of magnitude improvements in memory and compute efficiency. Our experiments isolate the impact of the rounding function in the quantization pipeline while varying transformations on a scaled min-max grid. Our results show that Qronos consistently offers improvements over previous state-of-the-art methods when quantizing weights, activations, and/or KV caches to 4 bits or fewer. That said, our results are intentionally limited to the scaled min-max quantization grid to focus our experiments on transformations and rounding; we believe our results could be further improved by leveraging weight and activation distributions to design quantization grids that are more effective than the scaled min-max grid used in this work, possibly with non-uniform grids via vector quantization.

## REPRODUCIBILITY STATEMENT

We integrate Qronos into the open source Brevitas quantization library. Our code and instructions on how to reproduce the various experiments, including hyperparameters and random seeds, can be found here on GitHub. The code points to all datasets and models used for calibration and evaluation.

## ACKNOWLEDGMENT

We gratefully acknowledge partial support by National Science Foundation, via the DMS-2410717 grant. We also gratefully acknowledge partial support from AMD. From AMD, we would like to thank Nick Fraser and Giuseppe Franco for their constructive reviews and feedback that made this paper better, and Gabor Sines, Michaela Blott, Syed Naim, Yonas Bedasso, and Max Kiehn for their support.

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

## A  PSEUDOCODE OF QRONOS

We provide the pseudocode for our efficient version of Qronos derived in Section 3.1.

---

**Algorithm 1** Qronos (Efficient Version)

---

$H = \widetilde{X}^\top \widetilde{X}, G = \widetilde{X}^\top X$

$H^{-1} = (\widetilde{X}^\top \widetilde{X})^{-1} = LL^\top$ ▷ Cholesky Decomposition

**for** every $w$ in $W$ (in parallel) **do**

$\quad q = \mathbf{0}^N$

$\quad w^{(0)} \leftarrow \text{copy}(w)$

$\quad q_1 = \mathcal{Q}\left(\dfrac{G_{1,\geq 1}w - H_{1,\geq 2}w_{\geq 2}^{(0)}}{H_{11}}\right)$ ▷ By Proposition E.2

$\quad w_{\geq 2}^{(1)} = L_{\geq 2,\geq 2}L_{\geq 2,\geq 2}^\top (G_{\geq 2,\geq 1}w - H_{\geq 2,1}q_1)$ ▷ By Lemma G.3

$\quad$ **for** $t = 2$ to $N$ **do** ▷ By Theorem 3.1 and Lemma 3.2

$\quad\quad q_t = \mathcal{Q}(w_t^{(t-1)})$

$\quad\quad w_{\geq t+1}^{(t)} = w_{\geq t+1}^{(t-1)} + \Delta^{(t)}$

$\quad\quad \Delta^{(t)} = -(w_t^{(t-1)} - q_t)\dfrac{L_{\geq t+1, t}}{L_{tt}}$

$\quad$ **end for**

**end for**

**return** $Q$

---

## B  RESULTS ON ADDITIONAL MODELS

Our main results evaluate Llama3 foundation models and Qwen3 instruction fine-tuned models. Here, we demonstrate that Qronos maintains the quality of Llama3 instruction fine-tuned models and Qwen3 foundation models as well. We again compare against RTN, OPTQ, GPFQ, and GPTAQ.

We present weight-only PTQ results with Llama3 instruction fine-tuned models at 3 and 4 bits in Table 6. As in Section 4.1, we asymmetrically quantize weights to the scaled min-max grid with $\beta = 1$ for both W3 and W4. We focus our instruction fine-tuned results on evaluating each rounding algorithm with and without Hadamard-based incoherence processing (HIP) as the quantization transform. As in Section 4.1, we find that HIP + Qronos consistently provides the highest quality quantized models relative to BF16 counterparts, as measured in both WikiText2 perplexity and zero-shot accuracy.

We then present weight-only PTQ results with Qwen3 foundation models in Table 7. We asymmetrically quantize weights to the scaled min-max grid with $\beta = 0.9$ for W3. We focus these results with and without Hadamard-based incoherence processing (HIP). Again, we find HIP + Qronos consistently yields the highest quality quantized models relative to the BF16 counterparts.

## C  EXPERIMENT DETAILS FOR QUANTIZATION TRANSFORMS

All experiments use WikiText2 as the calibration set, aside from SpinQuant, which uses C4. To pre-process our calibration dataset, we ensure that the `<bos>` token always appears as the first token in an input sequence as the recent study by Barbero et al. (2025) suggests removing `<bos>` during inference may greatly reduce performance if models were trained with `<bos>` always appearing at the first token; their analysis suggests the Llama3 family of models fits this category. Thus, to quantize our models, we first load the pre-trained checkpoint, then pre-process the dataset(s), then apply the quantization pipeline visualized in Figure 1. For Section 4.1, we intentionally select SmoothQuant (Xiao et al., 2023), Hadamard-based incoherence processing (HIP) (Ashkboos et al., 2024; Tseng et al., 2024), and MagR (Zhang et al., 2024) as they perform fundamentally different transformations. For Section 4.2, we study QuaRot, SmoothRot, and SpinQuant. Here, we describe hyperparameters for the data-driven transforms—SmoothQuant, MagR, SpinQuant, and SmoothRot.

Table 6: **Weight-only quantization of instruction fine-tuned Llama3 models.** We apply Hadamard-based incoherence processing (HIP) as our quantization transform (stage 1 in Figure 1) to isolate the impact of different rounding functions (stage 2) when quantizing to 3 and 4 bits, respectively denoted W3 and W4. We also evaluate no quantization transform (*i.e.*, "None").

| Stage 1 | Stage 2 | W3 | | | | | | W4 | | | | | |
| | | WikiText2 ($\downarrow$) | | | 0-shot ($\uparrow$) | | | WikiText2 ($\downarrow$) | | | 0-shot ($\uparrow$) | | |
| | | 1B | 3B | 8B | 1B | 3B | 8B | 1B | 3B | 8B | 1B | 3B | 8B |
|---|---|---|---|---|---|---|---|---|---|---|---|---|---|
| BF16 | - | 12.0 | 9.2 | 6.7 | 59.5 | 66.4 | 74.1 | 12.0 | 9.2 | 6.7 | 59.5 | 66.4 | 74.1 |
| None | RTN | 2e4 | 4e3 | 3e4 | 32.6 | 33.0 | 32.2 | 21.4 | 12.6 | 9.1 | 51.0 | 62.3 | 67.6 |
| | OPTQ | 60.0 | 16.1 | 12.2 | 37.4 | 49.9 | 58.2 | 14.3 | 9.9 | 7.3 | 54.5 | 63.6 | 71.8 |
| | GPFQ | 2e2 | 16.6 | 12.9 | 33.8 | 50.8 | 55.3 | 15.4 | 9.9 | 7.3 | 53.3 | 64.4 | 71.5 |
| | GPTAQ | 52.0 | 14.9 | 11.4 | 37.4 | 49.8 | 57.5 | **13.8** | 9.9 | 7.3 | **55.5** | 63.1 | 71.2 |
| | Qronos | **43.8** | **14.3** | **10.6** | **37.5** | **52.1** | **60.6** | **13.8** | 9.8 | 7.2 | **55.5** | 64.8 | 72.2 |
| HIP | RTN | 1e3 | 3e2 | 1e2 | 33.4 | 35.0 | 36.9 | 16.6 | 10.8 | 8.0 | 54.6 | 63.6 | 70.8 |
| | OPTQ | 19.1 | 12.8 | 9.3 | 48.0 | 58.2 | 59.0 | 13.2 | **9.6** | 7.1 | 56.6 | 64.5 | 72.1 |
| | GPFQ | 20.4 | 12.8 | 9.6 | 47.6 | 57.1 | 61.1 | 13.2 | 9.8 | 7.2 | 57.0 | **65.3** | 71.9 |
| | GPTAQ | 18.0 | 12.2 | 9.1 | 49.2 | 57.4 | 63.2 | 12.9 | 9.8 | **7.1** | 56.9 | 63.9 | **72.7** |
| | Qronos | **16.6** | **11.6** | **8.8** | **49.9** | **58.4** | **64.1** | **12.8** | **9.6** | **7.1** | **57.6** | 64.8 | 72.1 |

Table 7: **Weight-only quantization of Qwen3 foundation models to 3 bits with** $\beta = 0.9$**.** We apply Hadamard-based incoherence processing (HIP) as our quantization transform (stage 1 in Figure 1) to isolate the impact of different rounding functions (stage 2) when quantizing to 3 bits.

| Stage 1 | Stage 2 | WikiText2 ($\downarrow$) | | | 0-shot ($\uparrow$) | | |
| | | 1.7B | 4B | 8B | 1.7B | 4B | 8B |
|---|---|---|---|---|---|---|---|
| BF16 | - | 8.6 | 7.3 | 6.5 | 63.9 | 70.1 | 73.6 |
| None | RTN | 3e5 | 82.0 | 3e3 | 32.9 | 45.3 | 37.1 |
| | OPTQ | 37.5 | 10.4 | 8.8 | 35.7 | 57.2 | **61.7** |
| | GPFQ | 1e2 | 10.8 | 9.3 | 33.0 | 56.3 | 58.9 |
| | GPTAQ | 33.8 | 10.1 | 8.5 | **36.1** | **63.7** | 58.5 |
| | Qronos | **33.0** | **9.5** | **8.3** | 36.0 | 59.9 | 61.5 |
| HIP | RTN | 1e3 | 26.3 | 30.1 | 35.1 | 50.8 | 50.3 |
| | OPTQ | 10.8 | 8.8 | 7.6 | 54.4 | **64.4** | 67.6 |
| | GPFQ | 11.4 | 9.1 | 7.9 | 52.7 | 61.6 | 62.2 |
| | GPTAQ | 10.6 | 8.6 | 7.5 | 54.9 | 63.6 | 66.0 |
| | Qronos | **10.1** | **8.4** | **7.4** | **57.2** | 63.5 | **68.0** |

**SmoothQuant.** When applying SmoothQuant, we do so before quantizing weights or activations. In practice, SmoothQuant requires the selection of a hyperparameter to control the scaling optimization criteria. We refer to the SmoothQuant hyperparameter as $\gamma$ so as to not clash with our use of $\alpha$ in Section 4; note that $\gamma \in [0, 1]$. In Table 9, we provide the results of a uniform grid search over $\gamma$ when quantizing Llama3.2-1B-Instruct to 4 bits using round-to-nearest (RTN). These results motivate our decision to use $\gamma = 0.3$ in all our weight-only PTQ experiments that apply SmoothQuant.

**MagR.** When applying MagR, we also do so before quantizing weights and activations. When coupled with HIP, we do so after inserting rotations into the compute graph. In practice, MagR requires tuning the $\ell_\infty$ penalty; we refer to this hyperparameter as $\theta$, again so as to not clash with our use of $\alpha$ in Section 4. Zhang et al. (2024) tune $\theta$ to Llama2 models, settling on $\theta = 0.001$ for their experiments. In Table 10, we provide new results for Llama3.2-1B-Instruct. These results motivate our decision to use $\theta = 0.01$ in all our weight-only PTQ experiments that apply MagR.

**SpinQuant.** When applying SpinQuant, Liu et al. (2025) do so after activation (and KV cache) quantization but before weight quantization using an 800-sample calibration dataset; their ablation

Table 8: **3-bit weight-activation (W3A3) quantization of Llama3 foundation models.** We apply QuaRot as quantization transformation (stage 1) and compare different rounding functions (stage 2).

| | WikiText2 ($\downarrow$) | | | 0-shot ($\uparrow$) | | |
| | 1B | 3B | 8B | 1B | 3B | 8B |
|---|---|---|---|---|---|---|
| RTN | 2e3 | 9e2 | 1e3 | 33.0 | 32.3 | 32.8 |
| OPTQ | 9e2 | 2e2 | 1e2 | 32.3 | 33.2 | 35.9 |
| GPFQ | 60.0 | 30.1 | 27.9 | 35.6 | 39.2 | 40.3 |
| GPTAQ | 2e2 | 40.5 | 46.0 | 35.0 | 36.9 | 41.9 |
| Qronos | **46.8** | **22.0** | **20.4** | **37.0** | **43.4** | **47.4** |

Table 9: **Impact of SmoothQuant's $\gamma$ on Llama3.2-1B-Instruct.** We evaluate the impact of the smoothing parameter $\gamma$ on both WikiText2 perplexity and normalized average zero-shot accuracy when quantizing Llama3.2-1B-Instruct to 4 bits using round-to-nearest (RTN).

| $\gamma$ | 0.2 | 0.3 | 0.4 | 0.5 | 0.6 | 0.7 | 0.8 |
|---|---|---|---|---|---|---|---|
| **WikiText2** ($\downarrow$) | 24.6 | **18.6** | 18.9 | 21.4 | 87.0 | 4e2 | 3e4 |
| **0-shot** ($\uparrow$) | 50.8 | **53.3** | 52.8 | 52.5 | 42.8 | 36.6 | 32.3 |

study demonstrates negligible degradation when using 128 samples. Thus, we employ Cayley SGD on a network where only activations are quantized to optimize the learnable rotations for 100 iterations using a calibration dataset constructed of 128 random samples from the C4 dataset.

**SmoothRot.** When applying SmoothRot Czakó et al. (2025), we do so before quantizing weights or activations. Similar to SmoothQuant, SmoothRot requires the selection of a hyperparameter (*i.e.*, migration strength) to control the scaling optimization criteria. In our experiments, we use a migration strength of 0.6 as it empirically performed well for Llama3 1B.

## C.1 Grid scaling ablation study for 2 bits and fewer

In Section 4.1, we have presented weight-only PTQ results when quantizing to 2 bits or fewer on the scaled min-max grid with $\beta = 0.8$. Here, in Table 11, we provide additional results that demonstrate Qronos outperforms other rounding algorithms on another choice $\beta = 1$. Recall that we jointly apply Hadamard-based incoherence processing (HIP) and weight magnitude reduction (MagR) as quantization transforms before each rounding algorithm. Our results highlight that $\beta = 0.8$ (see Table 1) is an overall better choice for scaling the min-max grid in this setting, which is consistent with Zhang et al. (2024), and that Qronos provides the best results on both grids at all bit widths and model sizes. Our results with $\beta = 1$ also show that Qronos is more robust than GPTAQ when $\beta$ is not carefully selected.

## D More on Why Qronos Outperforms OPTQ

Let $W$ be the full-precision weights of a layer, and $Q$ their quantized counterparts. Let $X$ be the input to the layer and let its (possibly quantized) counterpart be $\widetilde{X}$; importantly, $\widetilde{X}$ reflects both activation quantization and the residual error propagated from previously quantized layers (possibly from previous blocks). Let $Y, \widetilde{Y}$ denote the respective outputs resulting from inputs $X, \widetilde{X}$.

For any single layer, OPTQ only attempts to minimize $\|\widetilde{X}(W - Q)\|_F$, which ignores the mismatch between $X$ and $\widetilde{X}$. In contrast, Qronos attempts to minimize $\|XW - \widetilde{X}Q\|_F$, which is the actual discrepancy between the full-precision outputs and their quantized counterparts.

A simple triangle inequality intuitively explains the distinction between OPTQ and Qronos:

$$\|Y - \widetilde{Y}\|_F = \|XW - \widetilde{X}Q\|_F \leq \|(X - \widetilde{X})W\|_F + \|\widetilde{X}(W - Q)\|_F.$$

While OPTQ only corrects the second term, Qronos corrects both terms. Thus, OPTQ only corrects quantization error in the weights at a given layer while Qronos corrects not only quantization error

Table 10: **Impact of MagR's $\theta$ on Llama3.2-1B-Instruct.** We evaluate the impact of the penalty parameter $\theta$ on both WikiText2 perplexity and normalized average zero-shot accuracy when quantizing Llama3.2-1B-Instruct to 4 bits using round-to-nearest (RTN).

| $\theta$ | 0.1 | 0.01 | 0.001 | 0.0001 |
|---|---|---|---|---|
| **WikiText2** ($\downarrow$) | 74.5 | **25.4** | 105.0 | 216.0 |
| **0-shot** ($\uparrow$) | 44.2 | **53.0** | 44.7 | 42.6 |

Table 11: **Weight-only quantization of Llama3 models to 2 bits or fewer with $\beta = 1$.** We jointly apply HIP and MagR as quantization transforms (stage 1 in Figure 1) and compare different rounding functions (stage 2) on the scaled min-max grid (see Section 4). Note that these results complement Table 1, which presents results with $\beta = 0.8$.

| | | **WikiText2** ($\downarrow$) | | | **0-shot** ($\uparrow$) | | |
|---|---|---|---|---|---|---|---|
| | | 1B | 3B | 8B | 1B | 3B | 8B |
| BF16 | - | 8.9 | 7.1 | 5.9 | 59.4 | 67.5 | 74.4 |
| | RTN | 1e4 | 1e4 | 2e4 | 32.4 | 32.4 | 32.9 |
| | OPTQ | 45.3 | 20.8 | 18.9 | 35.2 | 39.3 | 41.2 |
| 2-bit | GPFQ | 47.5 | 22.4 | 17.8 | 33.9 | 38.4 | 39.2 |
| | GPTAQ | 33.8 | 18.0 | 16.4 | 36.3 | 40.7 | 41.3 |
| | Qronos | **24.6** | **14.9** | **12.4** | **38.4** | **43.4** | **45.6** |
| | RTN | 2e5 | 3e5 | 6e5 | 32.0 | 32.6 | 32.1 |
| | OPTQ | 5e3 | 4e2 | 3e2 | 32.5 | 32.4 | 32.2 |
| 1.58-bit | GPFQ | 6e2 | 7e2 | 5e2 | 31.2 | 32.5 | 32.7 |
| | GPTAQ | 2e3 | 3e2 | 2e2 | 32.2 | 32.5 | 33.2 |
| | Qronos | **79.5** | **48.3** | **34.8** | **32.9** | **32.8** | **34.3** |

in both the weights and activations at a given layer, but also residual quantization error coming from previous layers, possibly from previous blocks.

Furthermore, tuning the quantization grid (*i.e.*, scaling factors and zeros points) cannot effectively minimize our objective in Equation 2. As before, decomposing $XW - \widetilde{X}Q = [(X - \widetilde{X})W] + [\widetilde{X}(W - Q)]$ isolates two error sources. Tuning quantization grids of the current layer only adjusts $Q$, and thus can affect only the second term, while the first term is untouched by any choice of quantization grids. Hence, it cannot close the performance gap between OPTQ and Qronos.

To illustrate this, we empirically compare quantization error accumulation by measuring the relative $\ell_2$ error, given by $\|Y - \widetilde{Y}\|/\|Y\|$, after each transformer block in Llama3.2 1B when quantizing weights to 3 bits, as in Section 4.1. Here, in Figure 3, we report the relative $\ell_2$ error averaged over each token in our calibration dataset (*i.e.*, 128 samples of 2048 tokens from WikiText2). Qronos yields the lowest average relative calibration error for each block, with 16% and 13% improvement over OPTQ and GPFQ, respectively, at the output of the final block.

# E  PRELIMINARY PROPOSITIONS

**Proposition E.1.** *The update rule given by*

$$q_t = \underset{p \in \mathcal{A}}{\arg\min} \frac{1}{2} \| Xw - \sum_{j=1}^{t-1} q_j \widetilde{X}_j - p \widetilde{X}_t - \sum_{j=t+1}^{N} w_j^{(t-1)} \widetilde{X}_j \|^2,$$

$$w_{\geq t+1}^{(t)} = \underset{(v_{t+1}, \dots, v_N) \in \mathbb{R}^{N-t}}{\arg\min} \frac{1}{2} \| Xw - \sum_{j=1}^{t} q_j \widetilde{X}_j - \sum_{j=t+1}^{N} v_j \widetilde{X}_j \|^2.$$

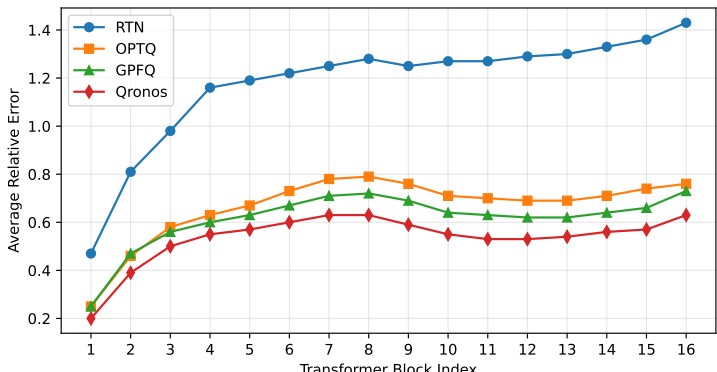

Figure 3: We visualize the evolution of the average relative error over transformer blocks when quantizing the Llama3 1B foundation model to 3 bits, further discussed in Appendix D.

*has closed-form expressions*

$$q_t = \mathcal{Q}\left(\frac{\langle Xw - \sum_{j=1}^{t-1} q_j \widetilde{X}_j - \sum_{j=t+1}^{N} w_j^{(t-1)} \widetilde{X}_j, \widetilde{X}_t \rangle}{\|\widetilde{X}_t\|^2}\right)$$

*and*

$$w_{\geq t+1}^{(t)} = \widetilde{X}_{\geq t+1}^{\dagger}\left(Xw - \widetilde{X}_{\leq t} q_{\leq t}\right).$$

*Proof.* For $q_t$, the corresponding optimization objective function is a one-dimensional quadratic function of $p$. Since minimizing a quadratic function over a discrete set $\mathcal{A}$ reduces to rounding its real-valued minimizer, we compute the real-valued minimizer

$$\frac{\langle Xw - \sum_{j=1}^{t-1} q_j \widetilde{X}_j - \sum_{j=t+1}^{N} w_j^{(t-1)} \widetilde{X}_j, \widetilde{X}_t \rangle}{\|\widetilde{X}_t\|^2}.$$

Thus, we obtain the closed-form expression of $q_t$,

$$q_t = \mathcal{Q}\left(\frac{\langle Xw - \sum_{j=1}^{t-1} q_j \widetilde{X}_j - \sum_{j=t+1}^{N} w_j^{(t-1)} \widetilde{X}_j, \widetilde{X}_t \rangle}{\|\widetilde{X}_t\|^2}\right),$$

where $\mathcal{Q}$ is the round-to-nearest operator.

For $w_{\geq t+1}^{(t)}$, the corresponding optimization problem is an unconstrained least-square problem in the form of $\min_{v \in \mathbb{R}^{N-t}} \|Ax - b\|^2$, with $A = \widetilde{X}_{\geq t+1}$ and $b = Xw - \widetilde{X}_{\leq t} q_{\leq t}$. Thus, the minimizer is given by $A^{\dagger}b$, which gives the desired closed-form expression. □

**Proposition E.2.** *The update rule given by*

$$q_1 = \mathcal{Q}\left(\frac{\widetilde{X}_1^{\top}(Xw - \widetilde{X}_{\geq 2} w_{\geq 2}^{(0)})}{\|\widetilde{X}_1\|^2}\right),$$

$$w_{\geq 2}^{(1)} = \widetilde{X}_{\geq 2}^{\dagger}\left(Xw - \widetilde{X}_1 q_1\right)$$

*is equivalent to*

$$q_1 = \mathcal{Q}\left(\frac{G_{1,\geq 1}w - H_{1,\geq 2}w_{\geq 2}^{(0)}}{H_{11}}\right)$$

$$w_{\geq 2}^{(1)} = (H_{\geq 2,\geq 2})^{-1}\left(G_{\geq 2,\geq 1}w - H_{\geq 2,1}q_1\right),$$

*where* $G = \widetilde{X}^T X \in \mathbb{R}^{N \times N}$ *and* $H = \widetilde{X}^T \widetilde{X} \in \mathbb{R}^{N \times N}$.

*Proof.* For $q_1$, we have $\widetilde{X}_1^\top X = (\widetilde{X}^\top X)_{1,\geq 1} = G_{1,\geq 1}$. Also, $\widetilde{X}_1^\top \widetilde{X}_{\geq 2} = (\widetilde{X}^\top \widetilde{X})_{1,\geq 2} = H_{1,\geq 2}$. Thus, $\widetilde{X}_1^\top (Xw - \widetilde{X}_{\geq 2}w_{\geq 2}^{(0)}) = G_{1,\geq 1}w - H_{1,\geq 2}w_{\geq 2}^{(0)}$. Further, $\|\widetilde{X}_1\|^2 = (\widetilde{X}^\top \widetilde{X})_{11} = H_{11}$. This gives the equivalence for updating $q_1$.

For $w_{\geq 2}^{(1)}$, $\widetilde{X}_{\geq 2}$ is given by $(\widetilde{X}_{\geq 2}^\top \widetilde{X}_{\geq 2})^{-1}\widetilde{X}_{\geq 2}^\top = (H_{\geq 2,\geq 2})^{-1}\widetilde{X}_{\geq 2}^\top$. Then

$$
\widetilde{X}_{\geq 2}^\dagger \left(Xw - \widetilde{X}_1 q_1\right) = (H_{\geq 2,\geq 2})^{-1}\widetilde{X}_{\geq 2}^\top \left(Xw - \widetilde{X}_1 q_1\right)
$$
$$
= (H_{\geq 2,\geq 2})^{-1}\left((\widetilde{X}^\top X)_{\geq 2,\geq 1}w - (\widetilde{X}^\top \widetilde{X})_{\geq 2,1}q_1\right)
$$
$$
= (H_{\geq 2,\geq 2})^{-1}\left(G_{\geq 2,\geq 1}w - H_{\geq 2,1}q_1\right).
$$

This gives the equivalence for updating $w_{\geq 2}^{(1)}$. $\qquad\qquad\qquad\qquad\qquad\qquad\qquad\square$

## F  PROOF OF THEOREM 3.1

*Proof.* We use induction to prove the theorem. Since at $t = 1$ equations Equation 3, Equation 4 and equations Equation 7, Equation 8 are identical, the base case is trivially true. Now we proceed with the induction, assuming $\hat{w}_{\geq t+1}^{(t)} = w_{\geq t+1}^{(t)}$ and $\hat{q}_t = q_t$.

Using definition Equation 3 and Proposition E.1, we can obtain the closed-form expression,

$$
q_{t+1} = \mathcal{Q}\left(\frac{\langle Xw - \sum_{j=1}^t q_j\widetilde{X}_j - \sum_{j=t+2}^N w_j^{(t)}\widetilde{X}_j, \widetilde{X}_{t+1}\rangle}{\|\widetilde{X}_{t+1}\|^2}\right),
$$

where $\mathcal{Q}$ is the RTN operator. Next we note that (4), which is used to compute $w_{\geq t+1}^{(t)}$, implies that $Xw - \sum_{j=1}^t q_j\widetilde{X}_j - \sum_{j=t+1}^N w_j^{(t)}\widetilde{X}_j$ is orthogonal to the column space of $\widetilde{X}_{\geq t+1}$. This in turn implies that $\langle Xw - \sum_{j=1}^t q_j\widetilde{X}_j - \sum_{j=t+1}^N w_j^{(t)}\widetilde{X}_j, \widetilde{X}_{t+1}\rangle = 0$. Then we can compute,

$$
q_{t+1} = \mathcal{Q}\left(\frac{\langle Xw - \sum_{j=1}^t q_j\widetilde{X}_j - \sum_{j=t+2}^N w_j^{(t)}\widetilde{X}_j, \widetilde{X}_{t+1}\rangle}{\|\widetilde{X}_{t+1}\|^2}\right)
$$
$$
= \mathcal{Q}\left(\frac{\langle Xw - \sum_{j=1}^t q_j\widetilde{X}_j - \sum_{j=t+1}^N w_j^{(t)}\widetilde{X}_j + w_{t+1}^{(t)}\widetilde{X}_{t+1}, \widetilde{X}_{t+1}\rangle}{\|\widetilde{X}_{t+1}\|^2}\right)
$$
$$
= \mathcal{Q}\left(\frac{\langle w_{t+1}^{(t)}\widetilde{X}_{t+1}, \widetilde{X}_{t+1}\rangle}{\|\widetilde{X}_{t+1}\|^2}\right)
$$
$$
= \mathcal{Q}\left(w_{t+1}^{(t)}\right) = \mathcal{Q}\left(\hat{w}_{t+1}^{(t)}\right) = \hat{q}_{t+1},
$$

where in the last two inequalities, we used the induction hypothesis $\hat{w}_{\geq t+1}^{(t)} = w_{\geq t+1}^{(t)}$ and the update rule (9).

Next, we prove $\hat{w}_{\geq t+2}^{(t+1)} = w_{\geq t+2}^{(t+1)}$. We first compute

$$
w_{\geq t+2}^{(t+1)} = \operatorname*{argmin}_{v_{\geq t+2}}\frac{1}{2}\|Xw - \sum_{j=1}^{t+1} q_j\widetilde{X}_j - \sum_{j=t+2}^N v_j\widetilde{X}_j\|^2
$$
$$
= \operatorname*{argmin}_{v_{\geq t+2}}\frac{1}{2}\|Xw - \sum_{j=1}^t q_j\widetilde{X}_j - \sum_{j=t+1}^N w_j^{(t)}\widetilde{X}_j + (w_{t+1}^{(t)} - q_{t+1})\widetilde{X}_{t+1} + \sum_{j=t+2}^N (w_j^{(t)} - v_j)\widetilde{X}_j\|^2.
$$

Due to the update rule (4), $Xw - \sum_{j=1}^{t} q_j \widetilde{X}_j - \sum_{j=t+1}^{N} w_j^{(t)} \widetilde{X}_j$ is orthogonal to the column span of $\widetilde{X}_{\geq t+1}$, hence to $(w_{t+1}^{(t)} - q_{t+1})\widetilde{X}_{t+1} + \sum_{j=t+2}^{N} (w_j^{(t)} - v_j)\widetilde{X}_j$. Then, we have

$$
w_{\geq t+2}^{(t+1)} = \underset{v_{\geq t+2}}{\mathrm{argmin}} \frac{1}{2} \| Xw - \sum_{j=1}^{t} q_j \widetilde{X}_j - \sum_{j=t+1}^{N} w_j^{(t)} \widetilde{X}_j + (w_{t+1}^{(t)} - q_{t+1})\widetilde{X}_{t+1} + \sum_{j=t+2}^{N} (w_j^{(t)} - v_j)\widetilde{X}_j \|^2
$$

$$
= \underset{v_{\geq t+2}}{\mathrm{argmin}} \frac{1}{2} \| (\hat{w}_{t+1}^{(t)} - \hat{q}_{t+1})\widetilde{X}_{t+1} + \sum_{j=t+2}^{N} (\hat{w}_j^{(t)} - v_j)\widetilde{X}_j \|^2 = \hat{w}_{\geq t+2}^{(t+1)},
$$

where we used the Pythagorean theorem, the induction hypothesis $\hat{w}_{\geq t+1}^{(t)} = w_{\geq t+1}^{(t)}$, and the fact $q_{t+1} = \hat{q}_{t+1}$. This completes the induction. $\qquad\square$

## G PROOF OF LEMMA 3.2

Throughout this section, we denote $H_{\geq t, \geq t} = X_{\geq t}^\top X_{\geq t} \in \mathbb{R}^{(N-t+1)\times(N-t+1)}$ and $H_{\geq t, \geq t}^{-1} = (X_{\geq t}^\top X_{\geq t})^{-1} \in \mathbb{R}^{(N-t+1)\times(N-t+1)}$. We will begin with a few preliminary lemmas before we prove Lemma 3.2. While some of these lemmas may already be known, we are not aware of any rigorous proofs in the literature. Thus, we provide our proofs here for completeness.

**Lemma G.1.** *Denote by $[H_{\geq t, \geq t}^{-1}]_{11}$ the first entry of $H_{\geq t, \geq t}^{-1}$ and by $[H_{\geq t, \geq t}^{-1}]_{\geq 2,1} \in \mathbb{R}^{N-t}$ the first column of $H_{\geq t, \geq t}^{-1}$ albeit with the first entry removed. Then*

$$
(X_{\geq t+1}^\top X_{\geq t+1})^{-1} X_{\geq t+1}^\top X_t = -\frac{[H_{\geq t, \geq t}^{-1}]_{\geq 2,1}}{[H_{\geq t, \geq t}^{-1}]_{11}}.
$$

*Proof.* We denote $r := [H_{\geq t, \geq t}^{-1}]_{11}$ and $\mathbf{b} = [H_{\geq t, \geq t}^{-1}]_{\geq 2,1}$. Then $\begin{pmatrix} r \\ \mathbf{b} \end{pmatrix}$ is just the first column of $H_{\geq t, \geq t}^{-1}$, so we have $H_{\geq t, \geq t} \begin{pmatrix} r \\ \mathbf{b} \end{pmatrix} = \mathbf{e_1}$. Let us write $H = \left[ \begin{array}{c|c} X_t^\top X_t & X_t^\top X_{\geq t+1} \\ \hline X_{\geq t+1}^\top X_t & X_{\geq t+1}^\top X_{\geq t+1} \end{array} \right]$. By comparing the two sides of $H \begin{pmatrix} r \\ \mathbf{b} \end{pmatrix} = \mathbf{e_1}$ we can observe $r X_{\geq t+1}^\top X_t + X_{\geq t+1}^\top X_{\geq t+1} \mathbf{b} = 0$, which implies

$$
(X_{\geq t+1}^\top X_{\geq t+1})^{-1} X_{\geq t+1}^\top X_t = -\frac{\mathbf{b}}{r}
$$

and finishes the proof. $\qquad\square$

The next lemma establishes how one can efficiently compute $H_{\geq t+1, \geq t+1}^{-1}$ from $H_{\geq t, \geq t}^{-1}$.

**Lemma G.2.** *$H_{\geq t+1, \geq t+1}^{-1}$ can be efficiently computed from $H_{\geq t, \geq t}^{-1}$ via*

$$
H_{\geq t+1, \geq t+1}^{-1} = \left( H_{\geq t, \geq t}^{-1} - \frac{1}{[H_{\geq t, \geq t}^{-1}]_{11}} [H_{\geq t, \geq t}^{-1}]_{\geq 1,1} [H_{\geq t, \geq t}^{-1}]_{1,\geq 1} \right)_{\geq 2, \geq 2}.
$$

We note that this is a simple rank-1 update followed by a submatrix slicing.

*Proof.* We first recall a more general inverse formula for $2 \times 2$ block matrix using the Schur complement. Consider the $2 \times 2$ block matrix

$$
M = \begin{pmatrix} A & B \\ C & D \end{pmatrix}.
$$

When $A$ is invertible, the inverse of $M$ is given by

$$
M^{-1} = \begin{pmatrix} A^{-1} + A^{-1}BS^{-1}CA^{-1} & -A^{-1}BS^{-1} \\ -S^{-1}CA^{-1} & S^{-1} \end{pmatrix}, \tag{19}
$$

where $S = D - CA^{-1}B$ is the Schur complement of $A$ in $M$.

When $A$ is a scalar $a$ and $M$ is symmetric, i.e.

$$M = \begin{pmatrix} a & b^\top \\ b & D \end{pmatrix},$$

this formula becomes

$$M^{-1} = \begin{pmatrix} a^{-1} + a^{-2}b^\top S^{-1}b & -a^{-1}b^\top S^{-1} \\ -a^{-1}S^{-1}b & S^{-1} \end{pmatrix},$$

where $S = D - a^{-1}bb^\top$.

By the Sherman–Morrison formula (Horn and Johnson, 2012), we have

$$D^{-1} = S^{-1} - \frac{S^{-1}bb^\top S^{-1}}{a + b^\top S^{-1}b}$$

$$= S^{-1} - \frac{a^{-2}S^{-1}bb^\top S^{-1}}{a^{-1} + a^{-2}b^\top S^{-1}b}.$$

Returning to our setting where $M^{-1} = H_{\geq t, \geq t}^{-1}$ and $D^{-1} = H_{\geq t+1, \geq t+1}^{-1}$, we have

$$H_{\geq t+1, \geq t+1}^{-1} = [H_{\geq t, \geq t}^{-1}]_{\geq 2, \geq 2} - \frac{1}{[H_{\geq t, \geq t}^{-1}]_{11}}[H_{\geq t, \geq t}^{-1}]_{\geq 2, 1}[H_{\geq t, \geq t}^{-1}]_{1, \geq 2}$$

$$= [H_{\geq t, \geq t}^{-1}]_{\geq 2, \geq 2} - \frac{1}{[H_{\geq t, \geq t}^{-1}]_{11}} \left( [H_{\geq t, \geq t}^{-1}]_{\geq 1, 1}[H_{\geq t, \geq t}^{-1}]_{1, \geq 1} \right)_{\geq 2, \geq 2}$$

$$= \left( H_{\geq t, \geq t}^{-1} - \frac{1}{[H_{\geq t, \geq t}^{-1}]_{11}}[H_{\geq t, \geq t}^{-1}]_{\geq 1, 1}[H_{\geq t, \geq t}^{-1}]_{1, \geq 1} \right)_{\geq 2, \geq 2}.$$

$\square$

Using the above lemma and Cholesky decomposition (Horn and Johnson, 2012), we can further simplify the right hand side in Lemma G.1 via the following lemma.

**Lemma G.3.** *Let $H^{-1} = (X^\top X)^{-1}$ and $H^{-1} = LL^\top$ be its Cholesky decomposition where $L$ is a lower triangular matrix, then*

$$\frac{[H_{\geq t, \geq t}^{-1}]_{\geq 2, 1}}{[H_{\geq t, \geq t}^{-1}]_{11}} = \frac{L_{\geq t+1, t}}{L_{tt}} \in \mathbb{R}^{N-t}$$

*holds for all $t \in [N-1]$.*

*Proof.* We first prove that given the Cholesky decomposition $H^{-1} = LL^\top$, the Cholesky decomposition of $H_{\geq t, \geq t}^{-1}$ is $H_{\geq t, \geq t}^{-1} = (L_{\geq t, \geq t})(L_{\geq t, \geq t})^\top$ for all $t \in [N]$, where $H_{\geq t, \geq t}^{-1} = (X_{\geq t}^\top X_{\geq t})^{-1} \in \mathbb{R}^{(N-t+1) \times (N-t+1)}$.

Let us proceed by induction. The base-case when $t = 1$ holds by assumption, and we now assume the result holds for $t$. By Lemma G.2, the updated inverse Hessian $H_{\geq t+1, \geq t+1}^{-1} = \left( H_{\geq t, \geq t}^{-1} - \frac{1}{[H_{\geq t, \geq t}^{-1}]_{11}}[H_{\geq t, \geq t}^{-1}]_{\geq 1, 1}[H_{\geq t, \geq t}^{-1}]_{1, \geq 1} \right)_{\geq 2, \geq 2}$. Thus,

$$\left( (L_{\geq t, \geq t})(L_{\geq t, \geq t})^\top - \frac{1}{L_{tt}^2}((L_{\geq t, \geq t})_{11} \cdot [L_{\geq t, \geq t}]_{\geq 1, 1})((L_{\geq t, \geq t})_{11} \cdot [L_{\geq t, \geq t}]_{\geq 1, 1})^\top \right)_{\geq 2, \geq 2}$$

$$= \left( (L_{\geq t, \geq t})(L_{\geq t, \geq t})^\top - [L_{\geq t, \geq t}]_{\geq 1, 1}[L_{\geq t, \geq t}]_{\geq 1, 1}^\top \right)_{\geq 2, \geq 2}$$

$$= ((L_{\geq t, \geq t})_{\geq 2, \geq 2})((L_{\geq t, \geq t})_{\geq 2, \geq 2})^\top$$

$$= (L_{\geq t+1, \geq t+1})(L_{\geq t+1, \geq t+1})^\top$$

This finishes the induction and we have Cholesky decomposition $H^{-1}_{\geq t, \geq t} = (L_{\geq t, \geq t})(L_{\geq t, \geq t})^\top$ for all $t \in [N]$. To finish the proof, let $M = RR^\top$ be the Cholesky decomposition of any positive definite matrix $M$. By a direct computation, the first column of $M$ is $R[R^\top]_{\geq 1,1} = R_{11} \cdot [R]_{\geq 1,1}$ and the first entry $M_{11} = R^2_{11}$. Then we have $\frac{M_{\geq 1,1}}{M_{11}} = \frac{[R]_{\geq 1,1}}{R_{11}}$ which implies that $\frac{M_{\geq 2,1}}{M_{11}} = \frac{[R]_{\geq 2,1}}{R_{11}}$. In our case, we have $H^{-1}_{\geq t, \geq t} = (L_{\geq t, \geq t})(L_{\geq t, \geq t})^\top$ in the place of $M = RR^\top$. Thus,

$$\frac{[H^{-1}_{\geq t, \geq t}]_{\geq 2,1}}{[H^{-1}_{\geq t, \geq t}]_{11}} = \frac{[L_{\geq t, \geq t}]_{\geq 2,1}}{[L_{\geq t, \geq t}]_{11}} = \frac{L_{\geq t+1, t}}{L_{tt}}.$$

$\square$

With the above preliminary lemmas, now we are ready to prove Lemma 3.2

**Proof of Lemma 3.2.** Since we initialize with $w^{(0)} = w$, $q_1 = \mathcal{Q}(w_1)$ always holds. Thus the two iterations produce the same $q_1$ and $w^{(0)}_{\geq 1}$. We proceed by induction. Assume at step $t$ that $q_t$ and $w^{(t-1)}_{\geq t}$ resulting from the update rules Equation 11 and Equation 12 match those following update rules Equation 13 and Equation 14. In order to complete the induction, it suffices to show that (12) and (14) produce the same $w^{(t)}_{\geq t+1}$, which naturally results in the same $q_{t+1} = \mathcal{Q}(w^{(t)}_{t+1})$.

To that end, we note that the optimization problem defined by Equation 12 has a unique least-square solution as $X_{\geq t+1}$ has full column rank. The minimizer is given by

$$w^{(t)}_{\geq t+1} = w^{(t-1)}_{\geq t+1} + (w^{(t-1)}_t - q_t)X^\dagger_{t+1:}X_t$$
$$= w_{\geq t+1} + (w^{(t-1)}_t - q_t)(X^\top_{\geq t+1}X_{\geq t+1})^{-1}X^\top_{\geq t+1}X_t$$

By Lemma G.1, we have

$$(X^\top_{\geq t+1}X_{\geq t+1})^{-1}X^\top_{\geq t+1}X_t = -\frac{[H^{-1}_{\geq t, \geq t}]_{\geq 2,1}}{[H^{-1}_{\geq t, \geq t}]_{11}}.$$

Lastly, Lemma G.3 gives us

$$\frac{[H^{-1}_{\geq t, \geq t}]_{\geq 2,1}}{[H^{-1}_{\geq t, \geq t}]_{11}} = \frac{L_{\geq t+1, t}}{L_{tt}} \in \mathbb{R}^{N-t}.$$

This matches $\Delta_{t+1}$ in Equation 14 and completes our induction. $\square$

## H    PROOF OF COROLLARY 3.4

---

**Algorithm 2** OPTQ: Quantize a layer $W$ given inverse Hessian $H^{-1} = (X^\top X)^{-1}$.

---
1: **for** every $w$ in $W$ in parallel **do**
2:     $q = \mathbf{0}^N$                                                    ▷ Initialize quantized neuron
3:     $H^{-1} = LL^\top$                                          ▷ Perform Cholesky decomposition
4:     **for** $t = 1$ to $N$ **do**                                            ▷ Iterate over rows
5:         $q_t = \mathcal{Q}(w_t)$
6:         $w_{\geq t} \leftarrow w_{\geq t} - L_{\geq t, t} \cdot (w_t - q_t)/L_{tt}$              ▷ Update remaining weights
7:     **end for**
8: **end for**
9: **return** $Q$

---

For our final result of this paper, we observe that updates of $w^{(t)}_{\geq t+1}$ via Equation 12 can be interpreted by observing that the term $(q_t - w^{(t-1)}_t)X_t$ represents the error introduced by quantizing $w^{(t-1)}_t$. The optimization problem Equation 12 seeks to mitigate this error by adjusting future weights so as to minimize the resulting distortion, measured in the $\ell_2$-norm. Notably, this step does not *explicitly*

attempt to correct errors introduced by earlier quantization steps $1, \ldots, t-1$. However, by combining the proof of Theorem 3.1 in the case when $X = \widetilde{X}$ with Lemma 3.2, we arrive at Corollary 3.4, which provides a novel interpretation of OPTQ. It shows—perhaps unexpectedly—that Algorithm 2 *optimally corrects* the cumulative weight quantization error incurred over the first $t$ entries of $w$.

*Proof.* The proof is based on induction on both arguments of the trajectory. Let $\{(\hat{w}^{(t-1)}_{\geq t}, \hat{q}_t)\}^N_{t=1}$ denote the trajectory generated by update rules Equation 17, Equation 18. And let $\{(w^{(t-1)}_{\geq t}, q_t)\}^N_{t=1}$ be the trajectory generated by Algorithm 2. Our goal is to prove $(\hat{w}^{(t-1)}_{\geq t}, \hat{q}_t) = (w^{(t-1)}_{\geq t}, q_t)$ for $t = 1, \ldots, N$.

By Lemma 3.2, the trajectory $\{(w^{(t-1)}_{\geq t}, q_t)\}^N_{t=1}$ generated using Cholesky decomposition in Algorithm 2 can be equivalently regarded as generated from Equation 11, Equation 12. Thus, we will use Equation 11, Equation 12 as the update rule of $w^{(t-1)}_{\geq t}$ and $q_t$ in the rest of our proof. In the base case, $\hat{w}^{(0)}_{\geq 1} = w^{(0)}_{\geq 1}$ are both initialized with $w$ and

$$\hat{q}_1 = \underset{p \in \mathcal{A}}{\operatorname{argmin}} \frac{1}{2} \|Xw - pX_1 - \sum_{j=2}^{N} w^{(0)}_j X_j\|^2 = \underset{p \in \mathcal{A}}{\operatorname{argmin}} \frac{1}{2}\|(w_1 - p)X_1\|^2 = \mathcal{Q}(w_1) = q_1.$$

Thus $(w^{(0)}_{\geq 1}, q_1) = (\hat{w}^{(0)}_{\geq 1}, \hat{q}_1)$. Assume $(\hat{w}^{(t-1)}_{\geq t}, \hat{q}_t) = (w^{(t-1)}_{\geq t}, q_t)$ holds true. Now we proceed to prove $(\hat{w}^{(t)}_{\geq t+1}, \hat{q}_{t+1}) = (w^{(t)}_{\geq t+1}, q_{t+1})$.

**Step 1:** We first prove $\hat{w}^{(t)}_{\geq t+1} = w^{(t)}_{\geq t+1}$. By construction,

$$\hat{w}^{(t)}_{\geq t+1} = \underset{v_{\geq t+1} \in \mathbb{R}^{N-t}}{\operatorname{argmin}} \frac{1}{2}\|Xw - \sum_{j=1}^{t} \hat{q}_j X_j - \sum_{j=t+1}^{N} v_j X_j\|^2.$$

For an arbitrary $v_{\geq t+1} \in \mathbb{R}^{N-t}$,

$$Xw - \sum_{j=1}^{t} \hat{q}_j X_j - \sum_{j=t+1}^{N} v_j X_j =$$

$$\underbrace{(Xw - \sum_{j=1}^{t-1} \hat{q}_j X_j - \sum_{j=t}^{N} \hat{w}^{(t-1)}_j X_j)}_{\textbf{(I)}} + \underbrace{\left((\hat{w}^{(t-1)}_t - \hat{q}_t)X_t + \sum_{j=t+1}^{N}(\hat{w}^{(t-1)}_j - v_j)X_j\right)}_{\textbf{(II)}}.$$

Since $\hat{w}^{(t-1)}_{\geq t+1}$ is a minimizer of Equation 18, the first term $\textbf{(I)} \in X^{\perp}_{\geq t}$, and clearly the second term $\textbf{(II)} \in \operatorname{span}\{X_t, \ldots, X_N\}$. Thus, we have

$$\left\|Xw - \sum_{j=1}^{t} \hat{q}_j X_j - \sum_{j=t+1}^{N} v_j X_j\right\|^2 = \|\textbf{(I)}\|^2 + \|\textbf{(II)}\|^2.$$

Notice that $\textbf{(I)}$ does not depend on $v_{\geq t+1}$. Furthermore, $\hat{w}^{(t-1)}_{\geq t+1}$ and $\hat{q}_t$ in $\textbf{(II)}$ can be replaced by $w^{(t-1)}_{\geq t+1}$ and $q_t$ respectively using our induction hypothesis. Thus,

$$\begin{aligned}
\hat{w}^{(t)}_{\geq t+1} &= \underset{v_{\geq t+1} \in \mathbb{R}^{N-t}}{\operatorname{argmin}} \frac{1}{2}\|Xw - \sum_{j=1}^{t} \hat{q}_j X_j - \sum_{j=t+1}^{N} v_j X_j\|^2 \\
&= \underset{v_{\geq t+1} \in \mathbb{R}^{N-t}}{\operatorname{argmin}} \frac{1}{2}\|(\hat{w}^{(t-1)}_t - \hat{q}_t)X_t + \sum_{j=t+1}^{N}(\hat{w}^{(t-1)}_j - v_j)X_j\|^2 \\
&= \underset{v_{\geq t+1} \in \mathbb{R}^{N-t}}{\operatorname{argmin}} \frac{1}{2}\|(w^{(t-1)}_t - q_t)X_t + \sum_{j=t+1}^{N}(w^{(t-1)}_j - v_j)X_j\|^2 \\
&= w^{(t)}_{\geq t+1}.
\end{aligned}$$

**Step 2:** Now we prove $\hat{q}_{t+1} = q_{t+1}$. We just constructed

$$\hat{w}_{\geq t+1}^{(t)} = \underset{v_{\geq t+1} \in \mathbb{R}^{N-t}}{\operatorname{argmin}} \frac{1}{2} \| Xw - \sum_{j=1}^{t} \hat{q}_j X_j - \sum_{j=t+1}^{N} v_j X_j \|^2.$$

This implies

$$Xw - \sum_{j=1}^{t} \hat{q}_j X_j - \sum_{j=t+1}^{N} \hat{w}_j^{(t)} X_j = P_{X_{\geq t+1}^{\perp}}(Xw - \sum_{j=1}^{t} \hat{q}_j X_j) \in X_{\geq t+1}^{\perp}. \tag{20}$$

By construction, we have

$$\hat{q}_{t+1} = \underset{q \in \mathcal{A}}{\operatorname{argmin}} \frac{1}{2} \| Xw - \sum_{j=1}^{t} \hat{q}_j X_j - q X_{t+1} - \sum_{j=t+2}^{N} \hat{w}_j^{(t)} X_j \|^2$$

$$= \mathcal{Q}\left( \frac{\langle X_{t+1}, Xw - \sum_{j=1}^{t} \hat{q}_j X_j - \sum_{j=t+2}^{N} \hat{w}_j^{(t)} X_j \rangle}{\|X_{t+1}\|^2} \right).$$

Then we can use Equation 20 to deduce

$$\frac{\langle X_{t+1}, Xw - \sum_{j=1}^{t} \hat{q}_j X_j - \sum_{j=t+2}^{N} \hat{w}_j^{(t)} X_j \rangle}{\|X_{t+1}\|^2}$$

$$= \frac{\langle X_{t+1}, Xw - \sum_{j=1}^{t} \hat{q}_j X_j - \sum_{j=t+1}^{N} \hat{w}_j^{(t)} X_j + X_{t+1} \hat{w}_{t+1}^{(t)} \rangle}{\|X_{t+1}\|^2}$$

$$= \frac{\langle X_{t+1}, Xw - \sum_{j=1}^{t} \hat{q}_j X_j - \sum_{j=t+1}^{N} \hat{w}_j^{(t)} X_j \rangle}{\|X_{t+1}\|^2} + \frac{\langle X_{t+1}, X_{t+1} \hat{w}_{t+1}^{(t)} \rangle}{\|X_{t+1}\|^2}$$

$$= \frac{\langle X_{t+1}, X_{t+1} \hat{w}_{t+1}^{(t)} \rangle}{\|X_{t+1}\|^2}$$

$$= \hat{w}_{t+1}^{(t)}$$

$$= w_{t+1}^{(t)}.$$

The last step $\hat{w}_{t+1}^{(t)} = w_{t+1}^{(t)}$ follows from what we just proved in Step 1 that $\hat{w}_{\geq t+1}^{(t)} = w_{\geq t+1}^{(t)}$. Thus we know

$$\hat{q}_{t+1} = \mathcal{Q}(\hat{w}_{t+1}^{(t)}) = \mathcal{Q}(w_{t+1}^{(t)}) = q_{t+1}.$$

This completes our induction. $\qquad\square$

