# OpenReview forum: "Qronos: Correcting the Past by Shaping the Future... in Post-Training Quantization"
_ICLR.cc/2026/Conference — ICLR 2026 Poster_

### Official Review · Reviewer_vydk · 2025-10-18

**Soundness:** 3
**Presentation:** 2
**Contribution:** 3
**Rating:** 4
**Confidence:** 3

**Summary:**

This paper studies post-training quantization (PTQ) problem and presents a method named Qronos for 4bits or lower. The key idea behind Qronos is to alternate between error correction and error diffusion via optimal update rules. Promising experimental results are reported for LLAMA3 and Qwen 3 models.

**Strengths:**

+ I have found the idea of alternating between error correction and diffusion novel and sensible.
+ The reported experimental results are promising and seem to be among the current SOTA.
+ The appendix contains detailed proofs for the theoretical results reported in Sec. 3.

**Weaknesses:**

- The literary presentation is still lacking. For example, "GPTAQ has been observed to be unstable in other reproductions." reads weird and the paper's introduction section needs to be revised significantly (too much emphasis on experimental results and little space for explaining the motivation or significance).
- Lemma 3.2 seems like a known result. I understand you provided rigorous proof for this lemma. But I believe that the connection between LS and Cholesky decomposition has been known in the literature.
- A minor weakness is the notable artifacts left by AI models (the notorious dash). For example, four dashes can be found on page 1 alone.

**Questions:**

1. Why did you name the proposed method Qronos? What is the purpose of adding "..." in the paper title?
2. Did authors compare Qronos with SmoothRot [1] at 4bit?
3. Is Theorem 3.1, Lemma 3.2, and Corollary 3.3 your own results or borrowed from the literature? If later, reference should be given.
[1] https://arxiv.org/pdf/2506.05413

---

> ### Author Response · Authors · 2025-11-25
>
> Thank you for your comments and questions. We have carefully addressed each of them, as outlined below, and have made revisions to the paper accordingly.
>
> > Did authors compare Qronos with SmoothRot [1] at 4bit?
>
> We thank the reviewer for bringing SmoothRot to our attention. We have now incorporated it as a stage 1 transformation option and added comprehensive results to Table 4. **Our updated table shows that the combination of Qronos and SmoothRot achieves particularly strong results, even for the challenging W4A4KV4 configuration.** Implementation details for the SmoothRot integration are now provided in Appendix C. We appreciate your suggestion, including SmoothRot strengthens our evaluation.
>
> > Lemma 3.2 seems like a known result. I understand you provided rigorous proof for this lemma. But I believe that the connection between LS and Cholesky decomposition has been known in the literature.
>
> > Is Theorem 3.1, Lemma 3.2, and Corollary 3.3 your own results or borrowed from the literature? If later, reference should be given.
>
> We thank the reviewer for this important clarification request. Theorem 3.1 and Corollary 3.3 are our original contributions and, to the best of our knowledge, novel results that have not appeared in existing literature. These constitute key theoretical contributions of our paper, as highlighted in the Introduction (page 1, lines 49-53) where we position them relative to concurrent work. Regarding Lemma 3.2, we explicitly state in footnote 2 that we do not claim novelty for this result. We included its proof for completeness because we could not find it stated in the precise form required for our framework. We would welcome any references the reviewer could provide for Lemma 3.2.
>
> > The literary presentation is still lacking. For example, "GPTAQ has been observed to be unstable in other reproductions." reads weird and the paper's introduction section needs to be revised significantly (too much emphasis on experimental results and little space for explaining the motivation or significance).
>
> We have made revisions to enhance clarity and readability throughout the paper. In particular, we have reworked the introduction to better articulate the motivation and significance of our method while keeping the experimental overview concise. Although Section 3.2 provides a detailed explanation of the motivation (as it requires formal notation and problem setup), the introduction now conveys these points more clearly to improve accessibility.
>
> Regarding the phrasing in the footnote, we understand that the original wording may have seemed unclear. Its intent was to explain the two NaN results in Table 4. After repeated occurrences of these NaNs, we verified that the official GPTAQ repository had reported issues related to instability during reproduction by others. We have rephrased the footnote to make this context explicit and improve clarity.
>
> > A minor weakness is the notable artifacts left by AI models (the notorious dash). For example, four dashes can be found on page 1 alone.
>
> We appreciate this observation. The frequent dashes reflect the writing style of one of our authors. Following your suggestion, we will revise the manuscript to vary our punctuation, replacing various dashes with commas and parentheses to improve readability.
>
> > Why did you name the proposed method Qronos? What is the purpose of adding "..." in the paper title?
>
> Thank you for asking about our naming choice. "Qronos" combines 'Q' for quantization with Chronos (Greek personification of time), reflecting how our algorithm corrects past quantization errors through future weight adjustments. The title was a stylistic choice.

---

### Official Review · Reviewer_4LxD · 2025-10-30

**Soundness:** 3
**Presentation:** 3
**Contribution:** 2
**Rating:** 4
**Confidence:** 5

**Summary:**

This paper investigates post-training quantization following a similar approach to the popular OPTQ/GPTQ framework. Specifically, the paper derives an optimization method that accounts for the accumulated errors from previously quantized layers. In addition, it proposes using Cholesky decomposition to accelerate the implementation. The method is evaluated on popular LLMs and combined with well-known quantization schemes, demonstrating improved accuracy in low-precision settings.

**Strengths:**

1). The paper is reasonably well written.

2). The intuition and analysis are solid, helping to clarify the motivation behind the proposed techniques as well as offering insights into the strengths and weaknesses of the widely used GPTQ method.

3). The experimental evaluation is comprehensive, particularly in combination with state-of-the-art techniques such as rotation and MagR.

4). The results generally show improved accuracy compared to previous methods.

**Weaknesses:**

1). The overall contribution is incremental.

2). The method introduces additional computational overhead to achieve accuracy gains over GPTQ. Since GPTQ also uses an approximate inverse Hessian to balance accuracy and speed, it is difficult to conclude that this method is definitively better than GPTQ.

3). The improvements are limited for 8B models, and no results are reported for very large models.

4). Some claims—particularly regarding implementation speedup and memory savings—seem misleading. The optimization process takes longer than GPTQ, yet the comparison is placed in the appendix. It would be more informative to present a detailed discussion of these trade-offs in the main paper to help assess the method’s advantages and limitations.

**Questions:**

1). The paper does not clearly explain how the algorithm is implemented. Could the authors include implementation details and pseudocode?

2). For the W4A4KV4 setting, the weight quantizer uses a linear search. How is this linear search integrated into the proposed optimization process?

---

> ### Author Response · Authors · 2025-11-25
> **Reply Part 1**
>
> Thank you for providing constructive feedback on our paper. Below, we carefully address each of your comments, and made revisions accordingly. We hope these changes resolve your concerns.
>
> > The improvements are limited for 8B models, and no results are reported for very large models.
>
> Thank you for raising this point. We have now added results in the main text (Table 2) for the Qwen3 model family ranging from 0.6B to 32B when quantizing their instruction fine-tuned checkpoints to 2 bits. Below, we provide WikiText2 perplexity for each model and method. **Qronos provides clear and consistent improvement over existing methods across model sizes.**
>
> **2-bit weight-only quantization of Qwen3 models (WikiText2 perplexity)**
> |   | 0.6B      | 1.7B      | 4B        | 8B       | 14B      | 32B      |
> |---------|-----------|-----------|-----------|----------|----------|----------|
> | **BF16**   | 18.6      | 15.2      | 12.2      | 8.6      | 7.6      | 6.8      |
> | **RTN**    | 6e5  | 7e6  | 4e5  | 4e4 | 3e5 | 1e5 |
> | **OPTQ**   | 148.0     | 60.0      | 22.8      | 14.7     | 14.9     | 12.8     |
> | **GPFQ**   | 105.0     | 45.3      | 25.4      | 17.1     | 15.6     | 13.4     |
> | **GPTAQ**  | 74.5      | 37.0      | 21.0      | 13.6     | 14.4     | 12.9     |
> | **Qronos** | **46.0**  | **23.5**  | **17.8**    | **12.9** | **13.4** | **12.0** |
>
> > Some claims—particularly regarding implementation speedup and memory savings—seem misleading. The optimization process takes longer than GPTQ, yet the comparison is placed in the appendix. It would be more informative to present a detailed discussion of these trade-offs in the main paper to help assess the method’s advantages and limitations.
>
> We appreciate your observation. Our original submission included a runtime analysis in the main text (Figure 2, page 9) to illustrate that our theoretical contributions substantially accelerate Qronos relative to its prior version, namely "Qronos (Base Ver.)" vs. "Qronos (Efficient Ver.)". We respectfully highlight that our analysis does intentionally separate the algorithm runtime (Figure 2a) from the overall runtime (Figure 2b) for transparency. While our theoretical insights in Section 3.1 enabled us to optimize the core computation loop of Qronos to scale comparably to OPTQ, this plot highlights that there is still a runtime overhead between Qronos and OPTQ.
>
> Since initial submission, we have optimized our software implementation (to be open-sourced) and conducted a thorough evaluation of end-to-end calibration on an AMD MI325X GPU. Below, we report end-to-end runtimes for OPTQ and Qronos when quantizing the Qwen3 model family (0.6B–32B), normalized to Qwen3-0.6B. The results show that **Qronos’s overhead decreases to just 8.7\% for the largest models, indicating that algorithmic cost dominates the cost of executing inference twice, and underscoring the importance of our theoretical insights**. In response to your feedback, we have moved the complete runtime analysis to the main text to provide a clearer and more balanced view of the method’s advantages and limitations.
>
> **End-to-end calibration time for Qwen3 models, normalized to Qwen3-0.6B**
> |  | 0.6B | 1.7B | 4B  | 8B  | 14B | 32B  |
> |------------|------|------|-----|-----|-----|------|
> | **OPTQ**   | 1.0  | 1.4  | 2.8 | 3.7 | 5.6 | 10.6 |
> | **Qronos** | 1.2  | 1.6  | 3.1 | 4.0 | 6.1 | 11.5 |
> | **Overhead** | 19.7\% | 16.2\% | 11.1\% | 9.0\% | 8.7\% | 8.7\% |

---

> > ### Author Response · Authors · 2025-11-25
> > **Reply Part 2**
> >
> > > The method introduces additional computational overhead to achieve accuracy gains over GPTQ. Since GPTQ also uses an approximate inverse Hessian to balance accuracy and speed, it is difficult to conclude that this method is definitively better than GPTQ.
> >
> > Thank you for the opportunity to clarify this point. **Qronos consistently outperforms OPTQ and other baselines across all evaluated settings.** Our results show clear and repeatable gains in model quality, making it difficult to justify choosing another algorithm given the evidence.
> >
> > In terms of efficiency, Qronos introduces only minimal overhead. From our new experiments, **calibration time increases by just 8.7\% for the largest model tested (i.e., Qwen3-32B)**. From a theoretical perspective, Appendix D explains why Qronos is fundamentally better than OPTQ: it corrects a major source of error that OPTQ cannot, while preserving compatibility with existing workflows. In short, Qronos costs very little extra in calibration time yet provides consistent benefits across multiple models and quantization configurations.
> >
> > > The overall contribution is incremental.
> >
> > Respectfully, we believe the assessment understates the scope of our contributions. Our work introduces both theoretical and algorithmic advances, supported by non-trivial proofs that are central to the paper. Specifically, **Qronos directly corrects a major source of error that OPTQ cannot address**, and our extensive experiments show that this correction significantly reduces end-to-end quantization error.
> >
> > We note that the connection between OPTQ and our efficient version of Qronos arose unexpectedly from our efforts to improve scalability. First, we derived Qronos from scratch using a principled objective that addresses a major source of error that OPTQ cannot (see Appendix D). Second, beyond efficiency gains, our analysis (Theorems 3.1 and Corollary 3.3) provides novel insight into OPTQ, showing that its seemingly local greedy updates actually progressively correct accumulated weight quantization error not only at the current time step, but from past steps. To our knowledge, this insight into the behavior of a widely used and highly cited algorithm is novel and not evident from its original formulation, its Cholesky-based implementations, or its predecessors like OBQ and OBS.
> >
> > Taken together, we believe the contribution of a principled algorithm with strong empirical results, coupled with new theoretical understanding of a widely used and highly cited method, goes well beyond incremental.

---

> > > ### Author Response · Authors · 2025-11-25
> > > **Reply Part 3**
> > >
> > > Thank you for your questions. Below, we carefully answer each of your questions. We have also made revisions accordingly.
> > >
> > > > The paper does not clearly explain how the algorithm is implemented. Could the authors include implementation details and pseudocode?
> > >
> > > We have now included pseudocode into Appendix A. We will also open source our efficient implementation if the paper is accepted.
> > >
> > > > For the W4A4KV4 setting, the weight quantizer uses a linear search. How is this linear search integrated into the proposed optimization process?
> > >
> > > For W4A4 and W4A4KV4 settings, we optimize the grid via a linear search between stages 1 and 2 of the quantization pipeline (see Figure 1). To be specific, we first perform the transformation (e.g., rotation), and then select the grid according to the transformed weights via linear search that minimizes the mean squared error between the weights and their quantized counterparts, i.e., $\Vert W - Q \Vert_F$.

---

### Official Review · Reviewer_uY12 · 2025-10-31

**Soundness:** 4
**Presentation:** 3
**Contribution:** 3
**Rating:** 8
**Confidence:** 5

**Summary:**

This paper introduces Qronos, a new post-training quantization (PTQ) algorithm for large language models that explicitly corrects errors accumulated across layers during quantization. Unlike previous methods such as OPTQ, which assume identical activations before and after quantization, Qronos accounts for the drift that occurs when earlier layers are already quantized. It alternates between two key steps: correcting current quantization errors and diffusing residual errors into future weights to prevent accumulation.
The authors provide a theoretical interpretation linking Qronos to OPTQ and derive an efficient implementation based on Cholesky updates. Extensive experiments are conducted to compare with other PTQ methods.

**Strengths:**

1. Introduces a clear mismatch-aware formulation addressing activation drift across layers.
2. Provides an efficient implementation for the proposed algorithm.
3. Shows consistent empirical gains over prior PTQ methods across various models and benchmarks.
4. Can be easily used with various quantization algorithms and improve their performance.
5. Offers useful theoretical insights linking and generalizing existing algorithms like OPTQ.

**Weaknesses:**

1. The proposed method mainly extends OPTQ rather than introducing a fundamentally new algorithmic framework.
2. The improvement on weight–activation quantization appears smaller than that on weight-only quantization; additional analysis would help clarify the underlying reason.
3. The main paper reports results primarily for 3- and 4-bit settings, where existing methods already perform well. Including 2-bit quantization experiments would better demonstrate the robustness of the proposed approach under more challenging conditions.

**Questions:**

See weakness

---

> ### Author Response · Authors · 2025-11-25
>
> Thank you for carefully reading our paper and providing a thoughtful review. Your comments on quantization difficulty (namely, weight-activation and 2-bit quantization) prompted us to conduct a deeper analysis that revealed an interesting pattern: **Qronos tends to provide larger gains on more challenging quantization tasks**.
>
> Specifically, we observe that while W4A4 shows smaller improvements than weight-only quantization at lower bit-widths (W3, W2, W1.58), the gains from Qronos actually increase as we add more quantization challenges. For instance, Qronos provides larger improvements on W4A4 compared to W4 weight-only, and even larger gains when we include KV cache quantization (W4A4KV4). **This suggests that our method is particularly effective when multiple sources of quantization error interact.**
>
> To further investigate this pattern, we have added W3A3 results to the appendix, which indeed show larger improvements than W4A4 and W3, supporting our hypothesis that Qronos excels on more difficult quantization scenarios. We have also added a brief discussion of this trend in Section 4.2, which we believe strengthens the paper's insights into when and why our method is most beneficial. We sincerely appreciate you pushing us to dig deeper into these results.
>
> To further strengthen our evaluation in response to your comments, we have now added a new Table 2 into the main text with 2-bit quantization results for the entire Qwen3 model family (0.6B to 32B) using instruction fine-tuned checkpoints. These new results complement Table 1 (section 1) in the main text and Table 11 (appendix C) in the appendix, which already includes comprehensive results for both 2-bit and 1.58-bit weight quantization on Llama3 models. Together, these results, with include both WikiText2 perplexity and 0-shot accuracy, confirm that Qronos maintains its effectiveness across different model architectures and sizes at 2-bit quantization, with especially notable improvements on smaller models where 2-bit quantization is most challenging.
>
> **2-bit quantization of Qwen3 model family (WikiText2 perplexity)**
> |   | 0.6B      | 1.7B      | 4B        | 8B       | 14B      | 32B      |
> |---------|-----------|-----------|-----------|----------|----------|----------|
> | **BF16**   | 18.6      | 15.2      | 12.2      | 8.6      | 7.6      | 6.8      |
> | **RTN**    | 6e5  | 7e6  | 4e5  | 4e4 | 3e5 | 1e5 |
> | **OPTQ**   | 148.0     | 60.0      | 22.8      | 14.7     | 14.9     | 12.8     |
> | **GPFQ**   | 105.0     | 45.3      | 25.4      | 17.1     | 15.6     | 13.4     |
> | **GPTAQ**  | 74.5      | 37.0      | 21.0      | 13.6     | 14.4     | 12.9     |
> | **Qronos** | **46.0**  | **23.5**  | **17.8**    | **12.9** | **13.4** | **12.0** |
>
> We appreciate you highlighting this aspect, as it reinforces one of our key findings: **Qronos provides the largest relative improvements precisely in these more challenging quantization scenarios where existing methods struggle most**.

---

### Author Response · Authors · 2025-11-25

We thank the reviewers for their thoughtful feedback. We are encouraged that the reviewers appreciated our framework for alternating between correction and diffusion, our theoretical analysis and its resulting insights, and our comprehensive experimentation showing consistent empirical gains over existing rounding methods. We also appreciate all of the constructive feedback the reviewers have provided, which we have carefully addressed with individual responses and corresponding changes to the paper. We believe the revisions have strengthened the paper. Below is a summary of the main changes we have made:

- Scaled our Qronos implementation to run efficiently on 32B models (Tables 2 and 5)
- Added 2-bit Qwen3 (0.6B-32B) results to the main text (Table 2)
- Expanded our discussion on calibration overhead and included Qwen3 (0.6B-32B) calibration time (Section 4.3)
- Added SmoothRot results to the main text (Table 4)
- Added W3A3 Llama3 experiments to the appendix (Table 8)

---

### Meta-Review · Area_Chair_UFxq · 2025-12-18

**Summary:**

Qronos is a post-training quantization algorithm that explicitly corrects errors from both weight and activation quantization, including errors accumulated from previously quantized layers, using an interpretable iterative framework that alternates error correction and diffusion. An equivalent formulation improves efficiency, reducing peak memory by 18× on Llama3-8B and achieving up to 13.8× single-layer speedups. Qronos is compatible with existing transforms and consistently outperforms prior adaptive rounding methods when quantizing weights, activations, and KV caches to 4 bits or fewer on Llama3 and Qwen3 models.


The reviewers' concerns include novelty (based on a limited baseline), performance on larger models, generality at lower bits, and computational overhead relative to the baseline. The AC thinks the authors provide a good rebuttal to these concerns. Therefore, I am inclined to accept this submission.

The ... style could easily be judged as LLM writing, and the AC thinks such details should be handled to avoid confusion. Moreover, the authors are encouraged to include these modifications in the final version.

**Reviewer Concerns:**

1. Reviewer uY12: Most concerns are solved; however, an in-depth analysis is helpful to fully address Weakness 2.
2. Reviewer 4LxD: Most concerns are solved except for the novelty issue. The AC thinks it's ok to address the novelty issue.
3. Reviewer vydk: Most concerns are solved.

**Reviewer Scores:**

1. Reviewer uY12 would keep the score as 8.
2. Reviewer 4LxD would maintain 4 or increase the score to 6.
3. Reviewer vydk would increase the score from 4 to 6.

---

### Decision · Program_Chairs · 2026-01-26

Accept (Poster)